# GENCOMPOSITOR: GENERATIVE VIDEO COMPOSITING WITH DIFFUSION TRANSFORMER

**Shuzhou Yang**[1]    **Xiaoyu Li**[2*]   **Xiaodong Cun**[4]    **Guangzhi Wang**[2]

**Lingen Li**[5]     **Ying Shan**[2]     **Jian Zhang**[1,3†]

[1]School of Electronic and Computer Engineering, Peking University     [2]ARC Lab, Tencent PCG

[3]Guangdong Provincial Key Laboratory of Ultra High Definition Immersive Media Technology

[4]GVC Lab, Great Bay University     [5]The Chinese University of Hong Kong

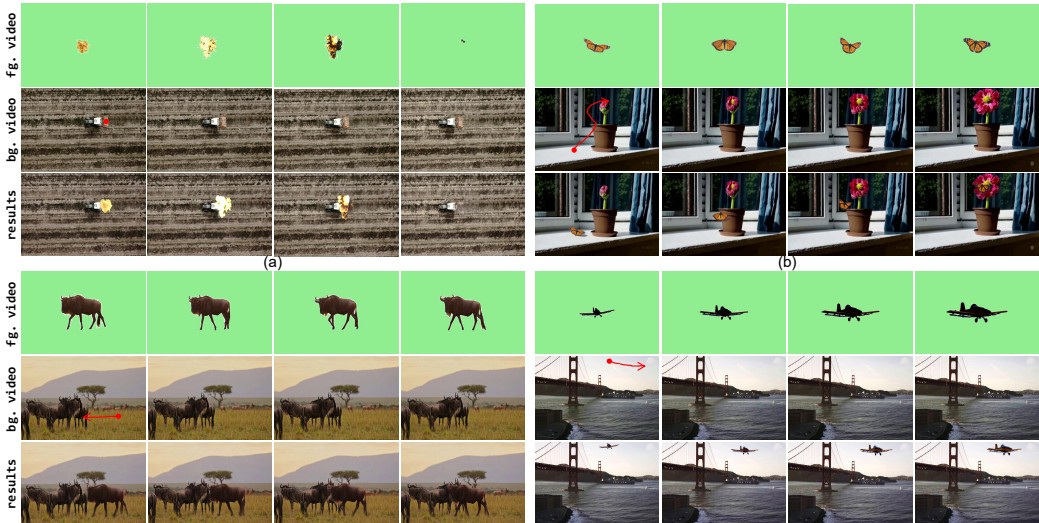

Figure 1: **GenCompositor** is capable of effortlessly compositing different videos following user-specified trajectories and scales. It could preserve the background video content and also seamlessly integrate the dynamic foreground elements into the background video, which not only strictly follows user-given instructions but also physically coordinates with background environments.

## ABSTRACT

Video compositing combines live-action footage to create video production, serving as a crucial technique in video creation and film production. Traditional pipelines require intensive labor efforts and expert collaboration, resulting in lengthy production cycles and high manpower costs. To address this issue, we automate this process with generative models, called generative video compositing. This new task strives to adaptively inject identity and motion information of foreground video to the target video in an interactive manner, allowing users to customize the size, motion trajectory, and other attributes of the dynamic elements added in final video. Specifically, we designed a novel Diffusion Transformer (DiT) pipeline based on its intrinsic properties. To maintain consistency of the target video before and after editing, we revised a light-weight DiT-based background preservation branch with masked token injection. As to inherit dynamic elements from other sources, a DiT fusion block is proposed using full self-attention, along with a simple yet effective foreground augmentation for training. Besides, for fusing background and foreground videos with different layouts based on user control, we developed a novel position embedding, named Extended Rotary Position Embedding (ERoPE). Finally, we curated a dataset comprising 61K sets of videos for our new task, called VideoComp. This data includes complete dynamic elements and high-quality target videos. Experiments demonstrate that our method effectively realizes generative video compositing, outperforming existing possible solutions in fidelity and consistency. Project is available at
https://gencompositor.github.io/

---

*Project leader.

†Corresponding author.

# 1 INTRODUCTION

Video compositing aims to edit target background video by adding foreground video from other sources, creating visually pleasing video assets. Compared to text- or image-guided video editing, this process edits video productions based on real-world captured videos, acting as a bridge between raw live-action video footage and final video work. However, classical process is labor intensive, which requires collaborative efforts of animators, videographers, and special effects artists. In this paper, we present generative video compositing, a new video editing task that attempts to automate compositing process with generative models (Ho et al., 2022; Blattmann et al., 2023; Li et al., 2025a). It allows direct editing of background videos using video footage under user control, aligning with traditional video creation process in video compositing stage.

To composite visually appealing video results, three main challenges are explored in this paper: 1) Ensuring background consistency of the video before and after editing. 2) Preserving the identity and motion of the injected dynamic elements while harmonizing with background content. 3) Enabling flexible user control, such as controllable size and motion trajectories. However, existing solutions cannot resolve these issues well. Controllable video generation (Hu, 2024; Li et al., 2026; Hu et al., 2025; Wang et al., 2025b) produces videos based on external conditions, including user-defined trajectories, images, and texts. Here, images provide identity information, and texts describe the motion to be generated. But images and text cannot precisely control the added elements at the pixel level, and current methods do not support external video condition. In Contrast, video harmonization methods (Lu et al., 2022; Chen et al., 2024; Xiao et al., 2024) directly paste foreground element onto the target video frame-by-frame, and adjust its RGB values to blend with background. However, these cannot adaptively specify the position, size, motion trajectory, and other attributes of the added element. Moreover, pasting foreground video requires accurate corresponding segmentation masks, which may not always be feasible in practical utilization.

To this end, we propose the first generative video compositing method, GenCompositor, aiming to automatically composite background video with dynamic foreground elements. As shown in Fig. 1, given a foreground video, a background video, and a user-provided trajectory in background (depicted by red line), GenCompositor injects foreground element to background video faithfully following trajectory. Trajectory can be specified by dragging a line or only clicking a point. For the latter, we track the movement of point with optical flow to simulate drag line. Our model can even predict realistic interaction between added element and background. For example, in Fig. 1 (a), the explosion effect influences the background content, *i.e.*, the car's fuel tank disappears after explosion in the last frame. In Fig. 1 (b), GenCompositor predicts realistic shadow of the added butterfly across frames, whose direction and effect consistent with background lighting.

We realize this by curating the first high-quality dataset for training, and proposing a novel Diffusion Transformer (DiT) pipeline specifically tailored for generative video compositing. Our pipeline consists of three main components. Firstly, a lightweight DiT-based background preservation branch is designed to ensure the consistency of the background in the edited results with the input videos. Secondly, to inject dynamic foreground elements, we propose a DiT fusion block with full self-attention to fuse tokens from foreground elements with that of background videos, and conduct detailed ablation study to demonstrate the advantages of this design over current cross-attention injection approaches (Ye et al., 2023; Cao et al., 2023; Wang et al., 2024a;c; Yang et al., 2025a). Considering that layouts of added foreground elements should follow user control and often differ from those of background videos, directly applying RoPE to foreground tokens introduces leakage artifacts. Therefore, we propose a novel position embedding method, Extended Rotary Position Embedding (ERoPE), to adaptively adjust the positions and scales of foreground tokens, which enables high-quality user-specified conditional generation even under layout-unaligned conditions. This technique is theoretically applicable to other tasks that exploit layout-unaligned video conditions.

Moreover, we develop some practical operations to enhance generalization and robustness. On the one hand, luminance augmentation is applied to foreground video for training, which strengthens the model's generalization to diverse foreground conditions. On the other hand, we propose mask inflation to feed inaccurate masks to the model, enabling realistic interaction between newly added objects and environment. Experiments show that GenCompositor enables high-quality video compositing and can be used to make video effects. Overall, our contributions are as follows:

- We propose a new practical video editing task, generative video compositing, and the first feasible solution that can automatically inject dynamic footage into the target video in a generative manner, utilizing the proposed novel diffusion architecture.

- We design some novel techniques targeted to the unique characteristics of video compositing, including a revised position embedding, a full self-attention DiT fusion block, and a lightweight background preservation branch. These provide references for future research.

- To train this model, we elaborate a data curation pipeline and develop some practical training operations, such as mask inflation and luminance augmentation, to improve the generalization ability and robustness of algorithm.

- Experiments prove that the proposed method outperforms existing potential solutions, enables effortless generative video compositing based on user-given instructions, and can be used for automatic video effects creation. This inspires future research in this area.

## 2 RELATED WORK

### 2.1 DIFFUSION-BASED VIDEO EDITING

In the field of AI-generated content, video editing (Brooks et al., 2023; Esser et al., 2023; Shin et al., 2024) has made great progress with the success of diffusion models (Yang et al., 2025d; Li et al., 2025b; Kong et al., 2024; Wang et al., 2025a; Ju et al., 2025; Li et al., 2025c). Early works attempted to use priors from pre-trained models (Ceylan et al., 2023; Yang et al., 2024; 2025b;c; Ma et al., 2025). Such as finetuning T2I models (Wu et al., 2023) or designing attention mechanisms tailored to video (Cai et al., 2025; QI et al., 2023; Wang et al., 2023; Yatim et al., 2025; Guo et al., 2024a). Recently, some methods trained specialized video editing models to realize more effective and prolific effect. Mou et al. (2024a) trained two ControlNet (Zhang et al., 2023) branches to specify motion and inject ID, respectively. Tu et al. (2025) followed a similar strategy, but enabled more fine-grained control through denser key points and trajectories. Liu et al. (2025) trained two control branches to guide spatial and temporal editing respectively. Bian et al. (2025) trained a video inpainting model, and edited videos through masking the target area, using Flux (Blackforestlabs, 2024) to edit the first frame, and painted subsequent frames to propagate the editing effect.

In general, training a dedicated model achieves better performance than training-free methods. However, existing approaches mainly edit videos based on images or textual prompts, making it hard to precisely control the appearance and motion details of the editing effects. To this end, we first propose generative video compositing task, which directly edits videos based on dynamic footage from other sources, realizing more practical and fine-grained video editing.

### 2.2 VIDEO HARMONIZATION

Video harmonization is a classical low-level vision task (Li et al., 2024; Dong et al., 2024; Tan et al., 2025), which aims to adjust the lighting of added foreground element in the composited video to make it visually harmonious. Similar to other low-level methods (Liu et al., 2022; Yang et al., 2023), Huang et al. (2020) first introduced adversarial training to video harmonization. Lu et al. (2022) collected a dedicated dataset for training. Ke et al. (2022) attempted to work in a white-box manner, which regresses the image-level filter argument to predict harmonized results. Guo et al. (2024b) directly trained a Triplet Transformer to simultaneously resolve multiple low-level video tasks, including harmonization. However, all of these methods focus only on adjusting the color of added elements. They all operate on composited videos and require a foreground video and its accurate and pixel-aligned mask as input. Besides, these methods do not allow users to freely modify the size and motion trajectory of the added elements in the final video. To address these limitations, we introduce generative video compositing task that accommodates arbitrary mask inputs. Using foreground videos, source background videos, and user-defined trajectories and scales, our model autonomously generates composited videos that adhere to all conditions.

## 3 TASK DEFINITION

Here, we define the task of generative video compositing. The inputs to this task are: a background video $\mathbf{v}_b$ to be edited, a foreground video $\mathbf{v}_f$ to be injected, and a user-given control $\mathbf{c}$. The objective is to composite $\mathbf{v}_b$ and $\mathbf{v}_f$ following $\mathbf{c}$, resulting in the final output $\mathbf{z}_0$.

Compared with other video editing methods, there are some unique characteristics of this problem. First, the layout of the videos to be edited $\mathbf{v}_b$ is highly consistent with editing results $\mathbf{z}_0$. Meanwhile, layouts of condition $\mathbf{v}_f$ are not pixel-aligned with $\mathbf{v}_b$. We should consider these two different conditions to generate results $\mathbf{z}_0$. Latent Diffusion Model (LDM) (Rombach et al., 2022) is employed to realize this, starting from a random latent noise $\mathbf{z}_T$, we aim to denoise $\mathbf{z}_T$ to $\mathbf{z}_0$ based on the three inputs mentioned above. For training, given the Ground Truth video $\mathbf{z}_0$ containing desired foreground element. We add noise $\epsilon$ of various scales to it following a predefined schedule defined as:

$$\mathbf{z}_t = \sqrt{\bar{\alpha}_t}\mathbf{z}_0 + \sqrt{1 - \bar{\alpha}_t}\epsilon. \tag{1}$$

Training process aims to predict the added noise $\epsilon$ in $\mathbf{z}_t$ by network $\epsilon_\theta(\cdot)$, objective function is:

$$\min_{\boldsymbol{\theta}} \mathbb{E}_{\mathbf{z}_0, \boldsymbol{\epsilon} \sim \mathcal{N}(0, I), t} \|\boldsymbol{\epsilon} - \epsilon_{\boldsymbol{\theta}}(\mathbf{z}_t | \mathbf{v}_b, \mathbf{v}_f, \mathbf{c}, t)\|_2^2, \tag{2}$$

where $t$ is sampling step, $\mathbf{v}_b$, $\mathbf{v}_f$, and $\mathbf{c}$ are conditions. After training, $\epsilon_\theta(\cdot)$ receives a random noise $\mathbf{z}_T$, denoises it step by step, and decodes final latent through a VAE decoder for composited results.

## 4 METHOD

We introduce input conversion in Sect. 4.1, showing how we convert and augment user-given instructions. Then, we explain the proposed pipeline, including background preservation branch (Sect. 4.2), DiT fusion block (Sect. 4.3), and Extended Rotary Position Embedding (ERoPE) (Sect. 4.4).

### 4.1 INPUT CONVERSION

For utilization, our inputs include a background video $\mathbf{v}_b$, a foreground footage $\mathbf{v}_f$, as well as the user-given scale factor $s$ and trajectory curve. We denote the user control as $u = \{\gamma, s\}$, where $\gamma$ is a 2D trajectory on the first frame of $\mathbf{v}_b$. We rasterize $u$ into a per-frame binary mask video $\mathbf{M} = \{\mathbf{M}_t\}_{t=1}^T$, $\mathbf{M}_t \in \{0, 1\}^{H \times W}$, and a masked video $\mathbf{X} = \{\mathbf{X}_t\}_{t=1}^T$ defined by

$$\mathbf{X}_t = (1 - \mathbf{M}_t) \odot \mathbf{v}_{b,t}, \quad t = 1, \dots, T, \tag{3}$$

where $\odot$ denotes element-wise multiplication. In Eq. 2, we thus set the conditioning variable to $\mathbf{c} = (\mathbf{M}, \mathbf{X})$, *i.e.*, $\epsilon_\theta(z_t \mid \mathbf{v}_b, \mathbf{v}_f, \mathbf{c}, t) = \epsilon_\theta(z_t \mid \mathbf{v}_b, \mathbf{v}_f, \mathbf{M}, \mathbf{X}, t)$. In this way, the user trajectory and scale are encoded entirely through the mask and masked background videos fed into the network.

As shown in the left of Fig. 2, our compositing starts from two input videos (background and foreground). Given a background video to be edited, users drag a trajectory on its first frame, denoting the movement of added elements. Alternatively, users can click a point (as shown in Fig. 1 (a)), and our method will automatically track the trajectory of this point in subsequent frames based on video optical flow. On this basis, the remaining problem is to determine the size of the newly added element. For the foreground input, we get its corresponding binary mask video through Grounded SAM2(Ren et al., 2024), then adjust its region size according to the user-given rescale factor. Finally, we adjust the position of this rescaled mask video based on the designated trajectory curve. In this way, we adaptively **rescale** and **reposition** foreground masks to produce a mask video.

To integrate with environment realistically, a Gaussian filter is used to inflate mask. Given the binary foreground mask video $\mathbf{M}^{\text{bin}} = \{\mathbf{M}_t^{\text{bin}}\}_{t=1}^T$, mask is inflated to define a buffered editable band around the object. We smooth $\mathbf{M}_t^{\text{bin}}$ with a 2D Gaussian kernel $G_\sigma$ and then apply a threshold:

$$\tilde{\mathbf{M}}_t = G_\sigma * \mathbf{M}_t^{\text{bin}}, \qquad \mathbf{M}_t = \mathbf{1}[\tilde{\mathbf{M}}_t > \tau], \quad t = 1, \dots, T, \tag{4}$$

where $*$ denotes convolution. Intuitively, $\mathbf{M}_t$ slightly extends the foreground region beyond the object boundary, creating a narrow buffer band in which the model is allowed to modify background content and hallucinate interactions (*e.g.*, shadows, glow, motion blur). At the same time, when SAM2 fails, this inflated mask explicitly absorbs pixel-level misalignments to support imperfect

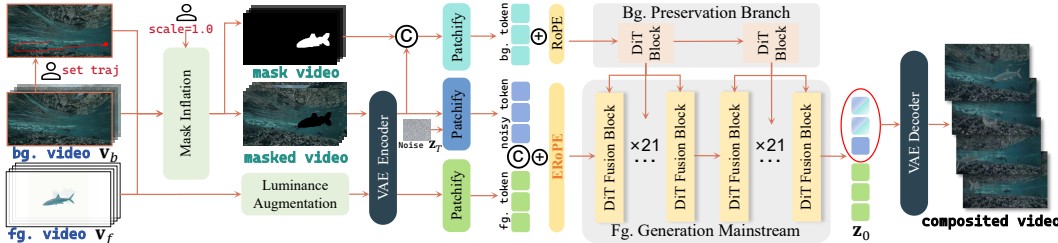

Figure 2: **Workflow of GenCompositor**. GenCompositor takes a background video and a foreground video as input. Users can specify the trajectory and scale of the added foreground elements. We firstly convert user-given instructions to model inputs, then generate with a background preservation branch and a foreground generation mainstream, consisting of the proposed ERoPE and DiT fusion blocks. Following this, our model automatically composites input videos.

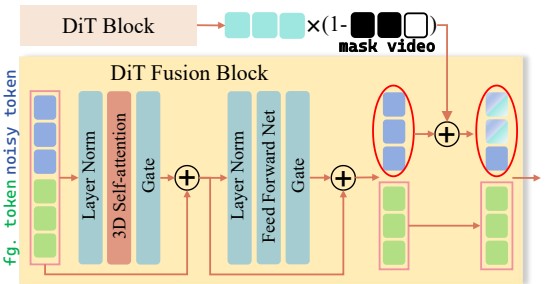

Figure 3: **DiT fusion block** receives concatenated tokens, some are to-be-generated tokens (blue) and some are unaligned conditional tokens (green). It fuses the two via pure self-attention. The final tokens with gradient color represent the mixture of generated tokens and masked conditional tokens from the BPBranch.

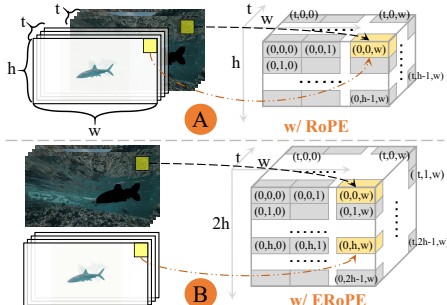

Figure 4: **RoPE and ERoPE.** (A) RoPE assigns labels to embeddings of videos. (B) For videos with inconsistent layout, our ERoPE extends RoPE by assigning unique labels to each embedding from different videos, avoiding interference.

masks. We also train our model with masks of SAM2, and the trained end-to-end model has a certain mask correction capability when utilization. Finally, masked video can be obtained by simply masking the object in the source video with a mask video for training. Our model actually receives (i) foreground video, (ii) inflated mask video, and (iii) masked video, to generate realistic output. This forces the model, during training, to explain pixels in the whole band around the object using both foreground and background cues, enabling local interaction.

## 4.2 BACKGROUND PRESERVATION BRANCH

As a video editing task, our main goal is to preserve non-editing content and only edit desired regions of $\mathbf{v}_b$. We propose a Background Preservation Branch (BPBranch) to ensure consistency between composited result and $\mathbf{v}_b$. Considering the overall layout of edited result is pixel-aligned with that of $\mathbf{v}_b$. Directly plus the latent of $\mathbf{v}_b$ with model latent has been able to faithfully preserve non-editing content, which is proven by previous ControlNet-like methods (Mou et al., 2024b; Ju et al., 2024).

Intuitively, only using masked video has been able to inject background. As shown in Fig. 2, in masked video, the area where we aim to insert the element is marked black. However, some background videos naturally contain black content, which is not the desired masks and could confuse the network. Hence, we concatenate the masked video with its corresponding mask video as input. This guides the branch to focus on background preservation.As mask video and masked video are pixel-aligned, we apply the same Rotary Position Embedding (RoPE) (Su et al., 2024) to both, as shown in Fig. 4(A). This strictly aligns the positions of the two and enables precise mask guidance.

Subsequently, we inject these to foreground generation mainstream for compositing. As BPBranch is designed to inject background only, its main purpose is to align features of masked video with that of mainstream model. Instead of deeply extracting masked features, we simply design a lightweight control branch, which consists of two normal DiT blocks, to align with the latent of mainstream. Meanwhile, as we only want to use the background region, a masked token injection method is

applied to prevent interference from BPBranch to foreground region, which is formulated as:

$$\mathbf{z}_t = \mathbf{z}_t + (1 - \mathbf{M}) \odot \mathbf{z}_{BPBranch}, \tag{5}$$

where $\mathbf{z}_t$ is the latent from mainstream, and $\mathbf{z}_{BPBranch}$ is the output of BPBranch. This process is visualized in detail in the upper part of Fig. 3.

### 4.3 FOREGROUND GENERATION MAINSTREAM

To composite foreground footage, our goal is to faithfully preserve the identity and dynamic features of foreground elements from other sources in composited results. Cross-attention is commonly used to inject conditions (FU et al., 2025; Yu et al., 2025), such as texts or camera poses. However, we find that although cross-attention could address semantic conditions, it does not effectively utilize low-level conditional information for our task. To faithfully inherit foreground condition, we propose to concatenate the tokens of foreground with the tokens to be denoised as shown in Fig. 2, then calculate its self-attention to fully fuse these two messages through the DiT fusion block.

As depicted in Fig. 3, given the tokens of noisy latent and foreground condition, DiT fusion block concatenates them in a token-wise manner, instead of classical channel-wise concatenation. This is because the layout features of foreground condition and generated results are not pixel-aligned. Roughly concatenating their features in channel dimension will cause severe content interference, which leads to training collapse. DiT fusion block then predicts the noise in noisy latent by calculating self-attention on the concatenated tokens, which contain both foreground condition and masked background. Note that our generated results are the processed latent that is boxed by red in Fig. 3. Hence, we fuse the tokens of BPBranch only with the processed noisy tokens and pass them to the next block. Finally, we only decode the part corresponding to the input noisy tokens to obtain composited video, as shown in the right of Fig. 2.

Meanwhile, newly added content should visually coordinate with background. Some of its attributes (*e.g.*, lighting), need to be properly adjusted during generation. To enable our model to learn this coordination adaptively, we develop a luminance augmentation strategy for training. In each iteration, we use gamma correction to the foreground video, with the gamma parameter randomly selected from a range of 0.4 to 1.9. This changes the lightness of foreground condition to offset it from source video. Consequently, based on our DiT fusion block that fully fuses foreground elements with model latent, foreground generation mainstream automatically learns the capabilities of foreground harmonization. Notice that luminance augmentation is only used in the training process.

### 4.4 EXTENDED ROTARY POSITION EMBEDDING

We distinguish two types of conditions with respect to the target layout. Given the background and target layout $(\mathbf{v}_b, \mathbf{z}_0)$, we call a condition video $\mathbf{v}$ *layout-aligned* if $\mathbf{v}(x, y)$ and $\mathbf{z}_0(x, y)$ refer to the same spatial location in the image plane, and *layout-unaligned* otherwise. In Fig. 2, the mask video $\mathbf{M}$ and masked video $\mathbf{X}$ are layout-aligned with $(\mathbf{v}_b, \mathbf{z}_0)$, while the foreground video $\mathbf{v}_f$ is layout-unaligned, since its dynamic content is centered and follows its own camera frame.

Layout-aligned conditions can be well utilized by sharing the same RoPE. However, this causes content interference for layout-unaligned conditions. As shown in Fig. 5, "w/ RoPE" directly shares RoPE between layout-aligned $(\mathbf{v}_b, \mathbf{M}, \mathbf{X})$ and layout-unaligned $\mathbf{v}_f$, which leads to obvious artifacts in composited results (boxed in red). Whose shapes and positions are consistent with that of foreground elements. Another optional way is cross-attention, which extracts abstract semantics to support layout-unaligned control signals. However,

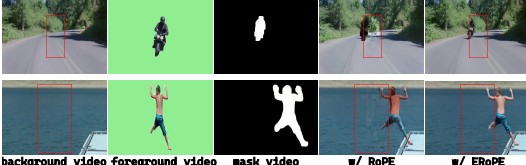

background video  foreground video  mask video  w/ RoPE  w/ ERoPE

Figure 5: ERoPE is superior in fusing layout-unaligned videos. As dynamic content in foreground footage is centered, using RoPE leads to content interference as shown in red box of "w/ RoPE", while our ERoPE resolves this issue well.

such a high-level feature cannot faithfully inherit detailed appearance and action features of foreground videos and is not suitable for video compositing, as proved in Sect. 5.4.

To this end, we developed a new embedding strategy tailored to the unaligned nature between foreground footage and background video, named ERoPE. It still takes the same four inputs

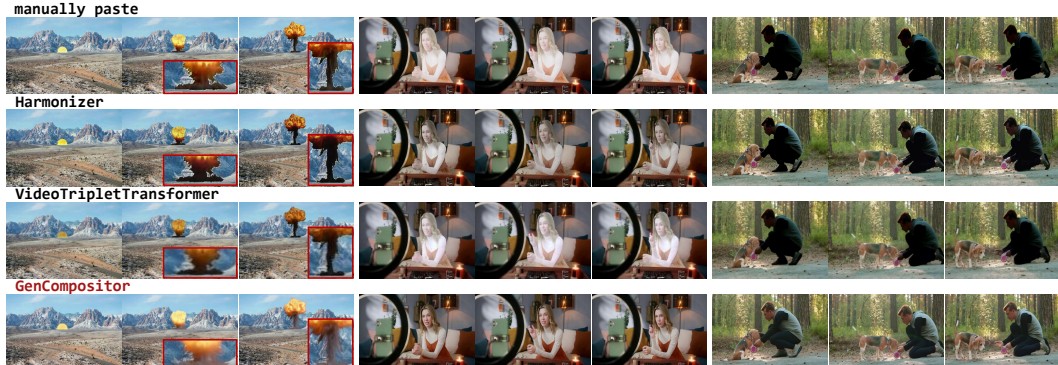

Figure 6: **Visual comparison of video harmonization**. The compared methods cannot achieve satisfactory results with jagged artifacts at the edges of foreground elements, inconsistent color or lighting, while our method achieves better performance.

$(\mathbf{v}_b, \mathbf{M}, \mathbf{X}, \mathbf{v}_f)$ as in Fig. 2, but changes the position embedding: we assign distinct positional labels to layout-unaligned tokens so that foreground and background videos can be fused without occupying the same spatial positions in the latent space. Concretely, in RoPE (Su et al., 2024), each token at 1D position index $p$ has its query/key components divided into 2D pairs and rotated by an angle proportional to $p$. For the $k$-th frequency $\omega_k$ and 2D query component $(q_{2k}, q_{2k+1})$, we have:

$$\begin{pmatrix} q'_{2k} \\ q'_{2k+1} \end{pmatrix} = \mathbf{R}(\theta_{p,k}) \begin{pmatrix} q_{2k} \\ q_{2k+1} \end{pmatrix}, \qquad \theta_{p,k} = \omega_k p, \quad \mathbf{R}(\theta) = \begin{pmatrix} \cos\theta & -\sin\theta \\ \sin\theta & \cos\theta \end{pmatrix}, \qquad (6)$$

and the same to the key component. Resulting attention score depends only on the relative position:

$$\langle q'_p, k'_q \rangle \;\propto\; f\big(\{\omega_k(p-q)\}_k\big), \qquad (7)$$

which implicitly assumes all tokens lie on a single spatio-temporal grid as shown in Fig. 4(A).

In our setting, tokens come from ForeGround and BackGround videos $s \in \{\text{BG}, \text{FG}\}$ with incompatible spatial coordinates. ERoPE introduces a stream-specific shift on top of RoPE:

$$\theta_{s,p,k} = \omega_k(p + \Delta_s), \qquad s \in \{\text{BG}, \text{FG}\}, \qquad (8)$$

where $\Delta_s$ is a constant offset for stream $s$. Let $q_{s,p} \in \mathbb{R}^D$ and $k_{s',q} \in \mathbb{R}^D$ denote the unrotated query and key for a token from stream $s$ at position $p$ and a token from stream $s'$ at position $q$, respectively. Here $s, s' \in \{\text{BG}, \text{FG}\}$ indicate the source streams of the query and key tokens. They may be equal for within-stream attention or different for cross-stream attention. We write $q'_{s,p}$ and $k'_{s',q}$ for their ERoPE-rotated versions obtained by applying $\mathbf{R}(\theta_{s,p,k})$ and $\mathbf{R}(\theta_{s',q,k})$ to each $(2k, 2k+1)$ pair. Using the identity $\mathbf{R}(\theta)^\top \mathbf{R}(\phi) = \mathbf{R}(\phi - \theta)$ and writing $\mathbf{q}_{s,p,k} = (q_{s,p,2k}, q_{s,p,2k+1})^\top$, $\mathbf{k}_{s',q,k} = (k_{s',q,2k}, k_{s',q,2k+1})^\top$ for 2D slices of $q_{s,p}$ and $k_{s',q}$, attention score between these two tokens is:

$$\begin{aligned} \langle q'_{s,p}, k'_{s',q} \rangle &= \sum_k \mathbf{q}_{s,p,k}^\top \mathbf{R}(\theta_{s',q,k} - \theta_{s,p,k}) \, \mathbf{k}_{s',q,k} \\ &= \sum_k \mathbf{q}_{s,p,k}^\top \mathbf{R}\big(\omega_k\big[(q + \Delta_{s'}) - (p + \Delta_s)\big]\big) \, \mathbf{k}_{s',q,k}. \end{aligned} \qquad (9)$$

In other words, the contribution of each frequency $k$ depends only on the relative effective positions $(p + \Delta_s)$ and $(q + \Delta_{s'})$, and hence on their difference $\omega_k[(p + \Delta_s) - (q + \Delta_{s'})]$. Choosing large, distinct shifts $\Delta_s$ for BG and FG prevents "same-position" collisions across streams, while still allowing attention to learn meaningful cross-stream interactions. As shown in Fig. 4(B), ERoPE is implemented as stream-specific shifts $\Delta_s$ on top of standard RoPE, introduces no additional parameters, and is crucial for stable fusion of layout-unaligned foreground and background videos in our compositing setting. As shown in Fig. 5, this strategy efficiently increases compositing performance and eliminates artifacts caused by interference of unaligned content. Moreover, considering that there are three optional shift directions for our ERoPE (e.g., width, height, and timing), this paper conducts ablation in Sect. F of appendix, and demonstrates their equality. We believe ERoPE is a practical tool for numerous layout-unaligned editing tasks, and here we use it for video compositing.

# 5 EXPERIMENTS

Given that there are no existing works for generative video compositing like ours, we compare with two related tasks, video harmonization, and trajectory-controlled video generation. We first introduce implementation details in Sect. 5.1, showcase comparisons in Sect. 5.2 and Sect. 5.3. To validate the effectiveness of our key components, we conduct an ablation study in Sect. 5.4.

Table 1: **Quantitative comparison** with video harmonization methods. The best results are highlighted in **bold**.

| Metrics | Harmonizer | VTT | GenCompositor |
|---------|-----------|-----|---------------|
| PSNR ↑ | 39.7558 | 40.0251 | **42.0010** |
| SSIM ↑ | 0.9402 | 0.9297 | **0.9487** |
| CLIP ↑ | 0.9614 | 0.9564 | **0.9713** |
| LPIPS ↓ | 0.0412 | 0.0455 | **0.0385** |

Table 2: **Quantitative results** of comparison with trajectory-controlled generation. The best results are highlighted in **bold**.

| Metrics | Tora | Revideo | VACE | GenCompositor |
|---------|------|---------|------|---------------|
| Subject ↑ | 88.44% | 88.02% | 89.51% | **89.75%** |
| Background ↑ | 92.45% | 92.90% | 92.63% | **93.43%** |
| Motion ↑ | 98.03% | 96.85% | 98.21% | **98.69%** |
| Aesthetic ↑ | 49.33% | 48.56% | 49.30% | **52.00%** |
| FVD ↓ | 1402.82 | 1342.56 | 942.52 | **535.71** |
| KVD ↓ | 94.94 | 64.53 | 120.92 | **45.91** |

## 5.1 IMPLEMENTATION DETAILS

GenCompositor is a DiT model with a 6B Transformer, consisting of the proposed DiT fusion blocks and background preservation branch. We reuse pre-trained VAE module of CogVideoX to generate in latent space. We train our new architecture on 8 H20 GPUs from scratch. In inference, GenCompositor takes over 65s to generate a video at $480 \times 720$ with 49 frames within 34GB VRAM.

## 5.2 COMPARISON WITH VIDEO HARMONIZATION

Considering the limited number of open-source methods in video harmonization, we compare our method with two recent approaches whose codes are available: Harmonizer (Ke et al., 2022) and VTT (Guo et al., 2024b). As these methods cannot control the motion trajectory, we only compare the ability to harmonize foreground elements. We also manually paste foreground footage from other sources into the background video for comparison. As shown in left of Fig. 6, noticeable jagged artifacts appear at the edges of added elements in manually paste and Harmonizer, due to the imperfection of the segmentation mask of the foreground video. Additionally, the color style of the newly added explosion effect does not harmonize well with background video in these harmonization approaches. In contrast, our method effectively addresses these jagged artifacts caused by inaccurate masks and produces more harmonious results. In other examples of Fig. 6, we manually adjust the lighting of the foreground elements and test the video harmonization ability. Our method consistently outperforms the other methods, demonstrating its superiority.

For quantitative comparison, we use the well-known HYoutube (Lu et al., 2022) dataset, where the foreground videos, segmentation masks, and source videos are available. Four well-known metrics, PSNR, SSIM (Wang et al., 2004), CLIP (Radford et al., 2021) and LPIPS (Zhang et al., 2018) are used to measure performance in Tab. 1. One can see that GenCompositor outperforms other harmonization methods across all metrics, showing its superiority over specialized methods in video harmonization. We also conduct a user study in Sect. E of appendix.

## 5.3 COMPARISON WITH CONTROLLABLE GENERATION

Our method enables trajectory-controlled generation, where users can specify the motion trajectory and size of newly added elements in results. Considering our added element is a dynamic video and there is no method using the same condition as ours, we compare with SOTA trajectory-controlled video generation and editing methods, Tora (Zhang et al., 2025), Revideo (Mou et al., 2024a), and VACE (Jiang et al., 2025). Visual results are given in Fig. 7, where we report background videos, foreground videos, Tora generation results containing the red trajectory curves, Revideo editing results, VACE generation, and our results. Note that Tora and VACE still require additional textual prompts as condition, Revideo and VACE have to edit the first frame as image condition, but GenCompositor requires neither of these priors. Although Tora and VACE could generate results that follow the trajectory, Tora cannot maintain the ID consistency of the added element and cannot strictly follow the user-specified trajectory. VACE gets object information only from the first-frame reference image, tends to mistake the object we wanted to inject and cannot predicts faithful dynamics. In contrast, GenCompositor could strictly follow the trajectories to generate composited videos,

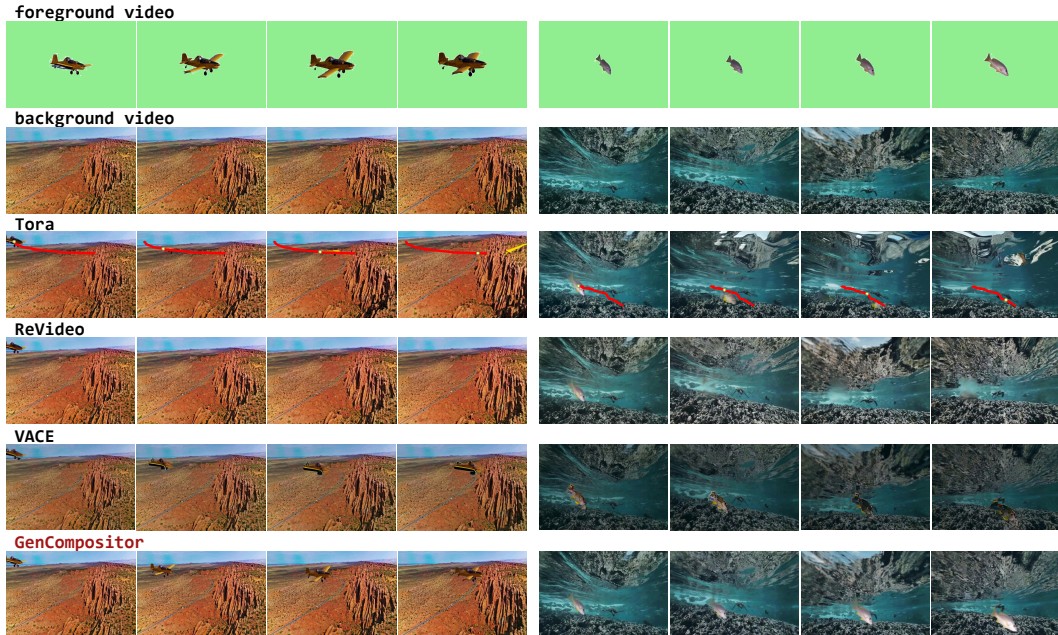

Figure 7: **Visual comparison of trajectory-controlled video generation**. All methods share the same trajectory, which is plotted as a red line in the results of Tora. All compared methods need to receive the pre-edited 1st-frame as condition image. Tora and Revideo generate temporally inconsistent foreground elements. VACE mistakes the element details we actually want to inject. Our method resolves these issues well without the preprocess of editing the first frame.

Table 3: **Quantitative results of ablation study.** The best results are highlighted in **bold**.

| Settings | PSNR ↑ | SSIM ↑ | CLIP ↑ | LPIPS ↓ | Subject ↑ | Background ↑ | Motion ↑ | Aesthetic ↑ |
|---|---|---|---|---|---|---|---|---|
| w/o fusion block | 19.8940 | 0.8015 | 0.9341 | 0.1535 | 88.85% | 92.21% | 98.34% | 48.85% |
| w/o BPBranch | 40.0099 | 0.9378 | 0.9709 | 0.0432 | 88.77% | 89.62% | 97.25% | 51.51% |
| w/o augmentation | 39.8040 | 0.9295 | 0.9629 | 0.0520 | 88.00% | 89.97% | 98.30% | 50.73% |
| w/o mask inflation | 41.8553 | 0.9422 | 0.9701 | 0.0409 | 89.72% | 91.62% | 98.28% | 50.87% |
| fullmodel | **42.0010** | **0.9487** | **0.9713** | **0.0385** | **89.75%** | **93.43%** | **98.69%** | **52.00%** |

and the ID and motion of the element are inherited from foreground videos faithfully. We believe these advantages come from different task settings. Compared with generating a video from an image such as Tora, or inpainting based on only single reference such as VACE, we composite foreground video following the trajectories. This is inherently more conducive to inheriting the ID and detailed motion of foreground elements. Meanwhile, Revideo also aims to drag the added element in the first frame to move along a given trajectory in subsequent frames, but its limited performance leads to non-robust results, elements may disappear in its predicted frames as shown in Fig. 7.

To conduct quantitative evaluations, we utilize two well-known generation quality metrics: FVD (Unterthiner et al., 2019) and KVD, and a common benchmark, *i.e.*, VBench (Huang et al., 2024; Zheng et al., 2025), to analyze the quality of generation from 4 dimensions: 1) Subject Consistency: consistency about subjects in the video. 2) Background Consistency: consistency about the video background. 3) Motion Smoothness: motion quality of the generated video. 4) Aesthetic Quality: subjective visual quality of the video. We believe these four metrics to be most relevant to our task and generate 40 sets of videos using Tora, Revideo, VACE, and our method, respectively. As shown in Tab. 2, our method achieves the best average scores across all six metrics. In addition, we conduct user study in Sect. E of appendix to compare intuitive quality of different methods.

## 5.4 ABLATION STUDY

To analyze effectiveness of each component, we conduct ablation study with four settings. Specifically, we attempt two potential designs of GenCompositor, one removes the background preservation branch (w/o BPBranch), the other uses cross-attention to inject foreground elements without the proposed DiT fusion block (w/o fusion block). We discuss their architecture details in Sect. G. We also remove two designs. One does not inflate binary mask (w/o mask inflation), the other directly inputs original foreground videos for training, without luminance augmentation (w/o augmentation).

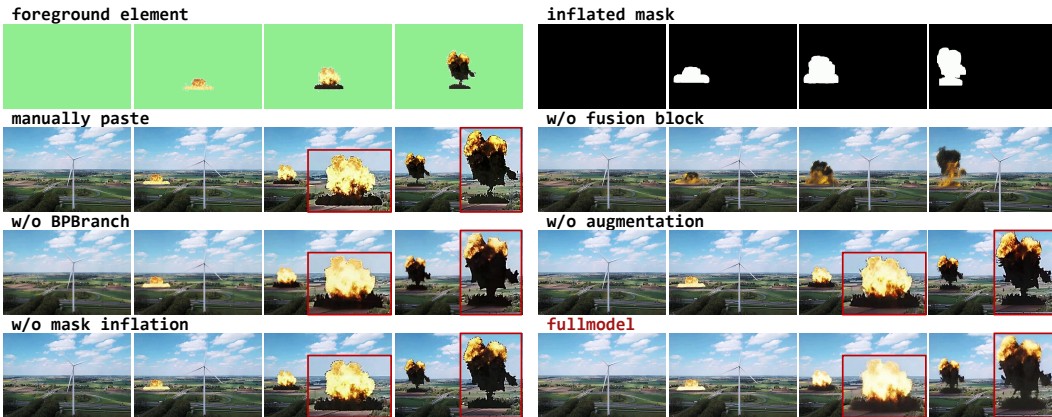

Figure 8: **Visual ablation results**. Manually paste results own jagged artifacts. "w/o fusion block" cannot inject element faithfully. "w/o augmentation" and "w/o mask inflation" showcase jagged artifacts at the edge of element. "w/o BPBranch" cannot realistically adjust added element. "fullmodel" setting performs best, which naturally fuses foreground elements with background video.

For visual comparison in Fig. 8, we provide the foreground element and inflated mask in the first row. Following three rows show the results of different settings. Where "manually paste", "w/o augmentation" and "w/o mask inflation" all exhibit obvious jagged artifacts at the edges of foreground elements. We believe that, for "w/o augmentation", due to the powerful learning ability of network, it totally inherits and overfits the content of the foreground video without any adjustment. For "w/o mask inflation", as we provide pixel-aligned mask to model, it can only process the foreground element in this limited region and cannot adjust surrounding pixels to fuse with the background, leaving artifacts at the edge. The other two settings address the border artifacts, but present other limitations. "w/o fusion block" cannot faithfully inject ID and dynamics of foreground elements, but successfully predicts the flame, which is semantically consistent with the foreground condition, indicating that cross-attention is good at injecting semantic information, but is not applicable in our task. Although "w/o BPBranch" also produces a realistic background, this end-to-end learning of both background video and added foreground elements increases training difficulty, limiting its performance. In its results, the foreground element is not perceptually consistent with main video.

For quantitative evaluation, we use four ablation settings, together with "fullmodel", to realize video harmonization and trajectory-controlled video generation, respectively. Related test data is the same as Sect. 5.2 and Sect. 5.3. As shown in Tab. 3, "fullmodel" outperforms all ablation settings on all metrics. We believe this objectively demonstrates the significance of each component in our method.

## 6 CONCLUSION

This paper introduces a novel video editing task, generative video compositing, which allows interactive video editing using dynamic visual elements. Specifically, we developed the first generative method, GenCompositor, which is designed to address three main challenges of this task: maintaining content consistency before and after editing, injecting video elements, and facilitating user control. It comprises three main contributions. Firstly, a lightweight background preservation branch is utilized to inject tokens of background videos into the mainstream. Secondly, the foreground generation mainstream incorporates novel DiT fusion blocks to effectively fuse the external video condition with the background latent. Finally, we revised a novel position embedding, ERoPE, to force the model to add external elements to the desired positions in results with desired scales, adaptively. Notice that ERoPE points out a new effective way to utilize layout-unaligned video conditions for generative model without any additional computational cost. The first paired dataset, VideoComp, is also proposed in this paper. Comprehensive experiments demonstrate the effectiveness and practicality of our proposed method.

**Acknowledgments.** This work was financially supported in part by National Natural Science Foundation of China (62372016), Guangdong Provincial Key Laboratory of Ultra High Definition Immersive Media Technology (2024B1212010006), Shenzhen Science and Technology Program (SYSPG20241211173440004) and Outstanding Talents Training Fund in Shenzhen.

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

CONTENTS

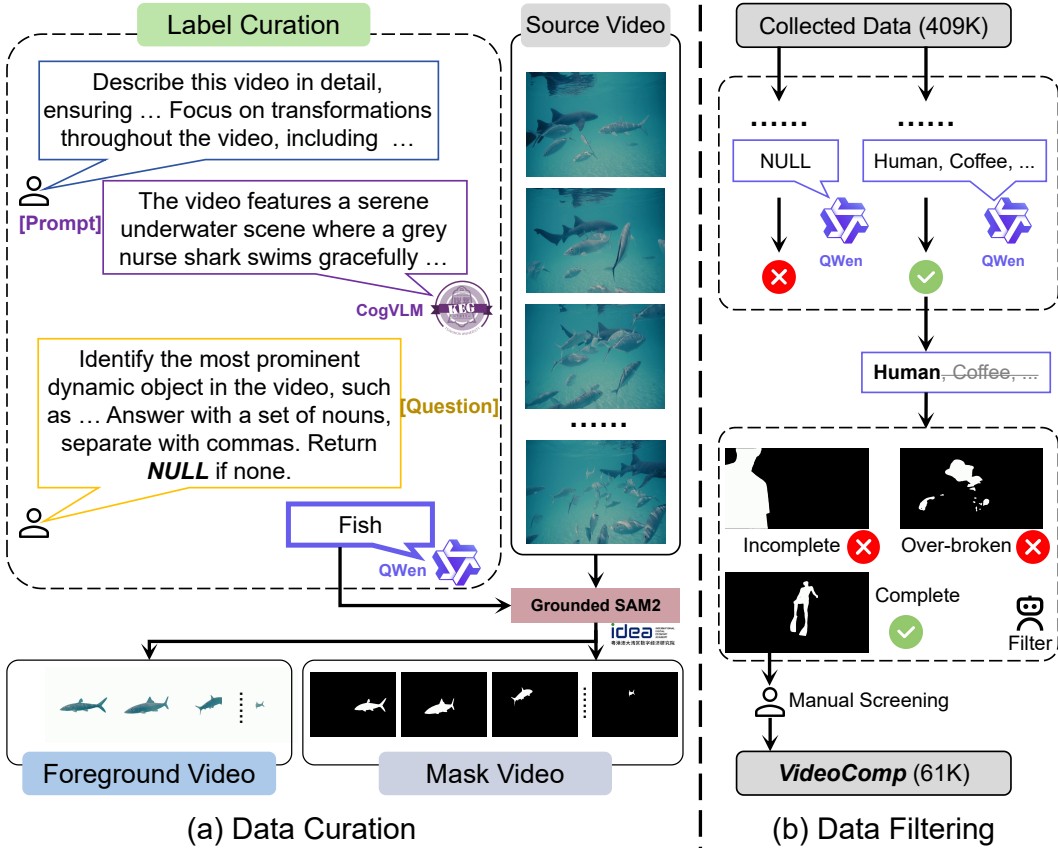

Figure 9: **Dataset construction pipeline**. We construct VideoComp dataset with two stages: data curation and data filtering. The former includes three steps: collection, labeling, and segmentation. Then, we select a high-quality subset based on several rules.

## A  STATEMENT

**Reproducibility statement.** We have provided anonymous code in supplementary materials. Due to file size limitations, model weights are not attached, but one can understand any details of our method in our code itself. For reproducibility, we will open-source all contributions of this paper when it is published, including code, model weights, and organized dataset. In addition, we provide a complete description of the data processing steps in Sect. B. The related code and the complete dataset will be open-sourced when this paper is published.

**Ethics statement.** This work does not involve human subjects, personal data, or any experiments that may raise ethical concerns. All datasets used are curated from internal source videos with appropriate licenses. The proposed methodology is intended for research purposes in video editing and does not pose foreseeable risks regarding fairness, privacy, or potential misuse. The authors have carefully reviewed and adhered to the ICLR Code of Ethics.

**The Use of Large Language Models (LLMs).** This paper did not involve the use of Large Language Models (LLMs) for research ideation, content generation, or writing. All content presented in this submission was created by the authors without significant contribution from LLMs. We confirm that we take full responsibility for the accuracy and integrity of the contents in this paper.

## B  DATASET CONSTRUCTION

Since there is no existing dataset for video compositing, we carefully build a dataset called VideoComp for this task. We present a scalable dataset construction pipeline that utilizes advanced models as tools for data construction (Wang et al., 2024b; Bai et al., 2023; Ren et al., 2024). The proposed VideoComp dataset contains 61K groups of videos, each containing three video samples, *i.e.*, the source video, the foreground video, and its corresponding mask video. The source video is a high-quality raw video and the other two are extracted

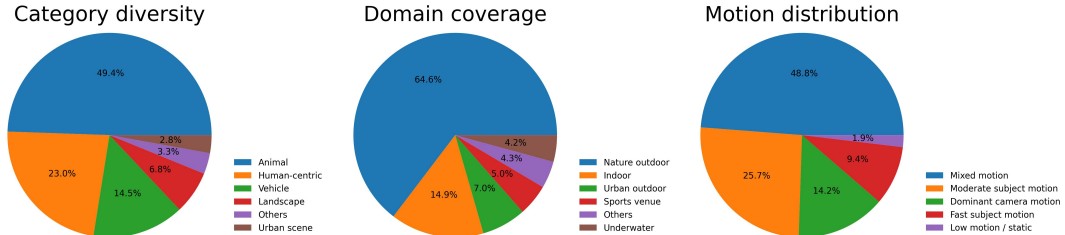

Figure 10: **Data type analysis**. We analyze data characteristics including category diversity, domain coverage, and motion distribution. Each is represented in a pie chart.

from it. As shown in Fig. 9, our dataset construction process consists of two main steps, data curation and data filtering.

### B.1 DATA CURATION

**(1)** We firstly collect about 240K cinematic HD videos that meet high aesthetic standards and large motion. Moreover, recently released Tiger200K (Zhou, 2025), which consists of 169K videos, is also included as our data source. For these total 409K videos, we propose an end-to-end workflow to process each of them as follows. **(2)** We distinguish the labels of prominent dynamic elements in the source video. We set up two questions to call CogVLM (Wang et al., 2024b) and QWen (Bai et al., 2023) respectively, where the former is a visual language model and the latter is a large language model. We input the video frames and then ask CogVLM to get the detailed description of video. Based on this answer and our second question, QWen is used to identify prominent dynamic elements, or return NULL if there are none. **(3)** Based on the output label of QWen, we employ Grounded SAM2 (Ren et al., 2024) to segment the elements in the video and save its foreground video and mask video. Note that the mask video shows the original motion trajectory. However, when saving the foreground video, *we center the dynamic element in each frame, eliminating its global position and trajectory information*. This approach allows us to control the trajectory of the resulting video totally based on the mask video, rather than relying on the position in the foreground video.

### B.2 DATA FILTERING

To ensure a high-quality dataset construction, we filter out unsatisfactory cases according to several rules. As illustrated in right of Fig. 9, our filtering principles are as follows: First, we exclude cases where QWen returns NULL, indicating that there is no significant object in the video. Second, for videos containing multiple elements, we select only the element with the highest probability. Third, we exclude suboptimal cases where elements have incomplete or excessively fragmented structures. Finally, we manually filter out videos that are visually unappealing.

### B.3 DATA TYPE ANALYSIS

To ease utilization, here we showcase three pie chats to analyze our VideoComp from three perspectives respectively, including category diversity, domain coverage, and motion distribution. As shown in Fig. 10.

(1) Category diversity: we list 5 categories of topics. Where half (49.4%) of the videos feature various animals, and human subjects account for almost a quarter (23.0%). Various vehicles account for 14.5%, including plane, car, and bus. 6.8% of videos record landscape , and urban scene occupies 2.8%. The remaining 3.3% is about complex topics.

(2) Domain coverage: we analyze the types of scenes displayed in the video data. The majority of these are nature outdoor pieces (64.6%), which typically offer a wider field of view and incorporate various real-world physical interactions and natural movements. This indirectly demonstrates the high quality of VideoComp. Indoor scenes account for 14.9%, urban outdoor accounts for 7.0%, and sports venue taks about 5.0%. The remaining 4.2% is about underwater scenes, and the other 4.3% includes various other scenes.

(3) Motion distribution: we also list the type of motion in the right end of Fig. 10. Where extreme motion only occupies 9.4% and 1.9%, for fast and low motion, respectively. Over a quarter of videos (25.7%) contain moderate subject motion, and almost a half of videos (48.8%) actually have mixed motion, which include multiple subjects with different amplitudes of motion. These strongly support the validity of the content in the proposed VideoComp dataset. In additiion, our data also contains approximately 14.2% camera motion video, which makes the model practical in real application.

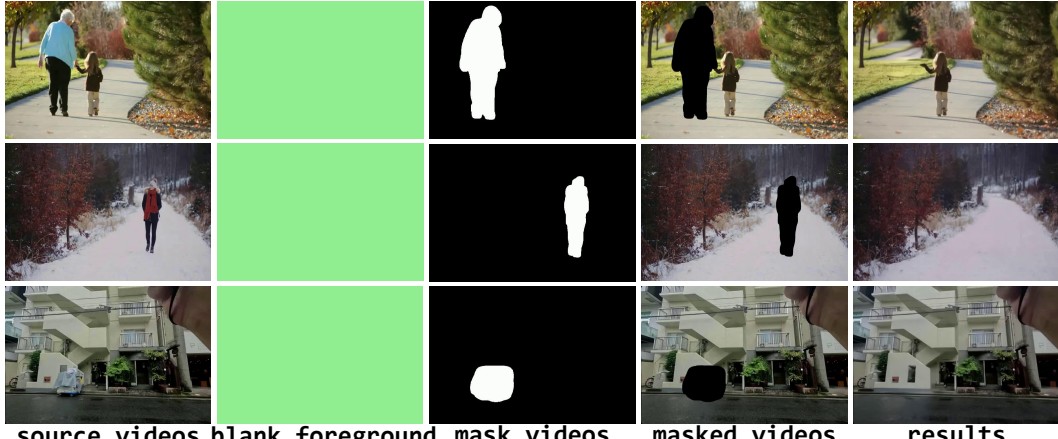

source videos blank foreground mask videos masked videos results

Figure 11: **Generalizability**. After training for video compositing, our method seamlessly enables video inpainting and the removal of target objects from videos with blank foreground condition.

## C   GENERALIZATION ABILITY

In addition to the effective video compositing capability of the proposed GenCompositor, our method also demonstrates impressive generalizability in video inpainting and the removal of target objects from videos, which can be achieved by simply replacing the foreground condition with a blank foreground video.

As shown in Fig. 11, we employ a blank foreground condition to remove target objects in source videos through the trained GenConpositor. We first use SAM2 (Ravi et al., 2025) to obtain mask videos corresponding to the objects to be removed in source videos. Then, as mentioned in Sect. 4.1, the same mask inflation operation is applied to get the inflated mask videos and masked videos. Given the blank foreground condition, GenCompositor removes objects and paints the masked region well. This successful generalization study demonstrates the significance of our proposed generative video compositing task, *i.e.*, this new video editing task inherently supports other video-related downstream tasks.

## D   IMPLEMENTATION DETAILS

We refer to CogVideoX-I2V-5B to design our model. Similarly to other DiT models, our GenCompositor also consists of 3 main components, Transformer, VAE, and text encoder, where we inherit the pre-trained weights of VAE and text encoder of CogVideoX in our model and do not train them. We only train Transformer here. Notice that as we provide null text during training, although GenCompositor inherits a text encoder, for inference, video compositing is totally based on the input videos and user-specified control, which is consistent with classical handmade process. The number of parameters of each component is: VAE (215,583,907), text encoder (4,762,310,656), Transformer (5,872,247,936), totaling 10,850,142,499. Following previous work, we only consider parameter number of Transformer in latent diffusion model, and claim that GenCompositor is a 6B model.

Our Diffusion Transformer contains two branches, one background preservation branch containing 301,568,576 parameters, and one foreground generation mainstream with 5,570,679,360 parameters. Since the original CogVideoX-I2V-5B is designed for image-to-video generation, we revised a novel module (DiT fusion block) to meet the characteristics of our task, video compositing, and train this new model from scratch. The foreground generation mainstream consists of 42 DiT fusion blocks.

As shown in Fig. 2 in main paper, our new DiT model has three patchify modules, which aim to patchify the features of input videos encoded by the VAE encoder into tokens that can be processed by Transformer, where the features encoded by VAE encoder contains 16 feature channels. Notice that for the background preservation branch, its inputs are a mask video and a masked video, which are concatenated on channel dimension. Hence, the input channel number of its patchify is 32. For foreground generation mainstream, its inputs are masked video, noise input, and foreground video condition. During training, the noise input is the combination of random noise and masked video, similar to other diffusion models. For inference, the noise input is pure gaussian noise. Notice that noise input is concatenate with masked video in channel dimension. Hence the input channel number of blue patchify in Fig. 2 is 32, and the input channel number of green patchify for foreground video condition is 16.

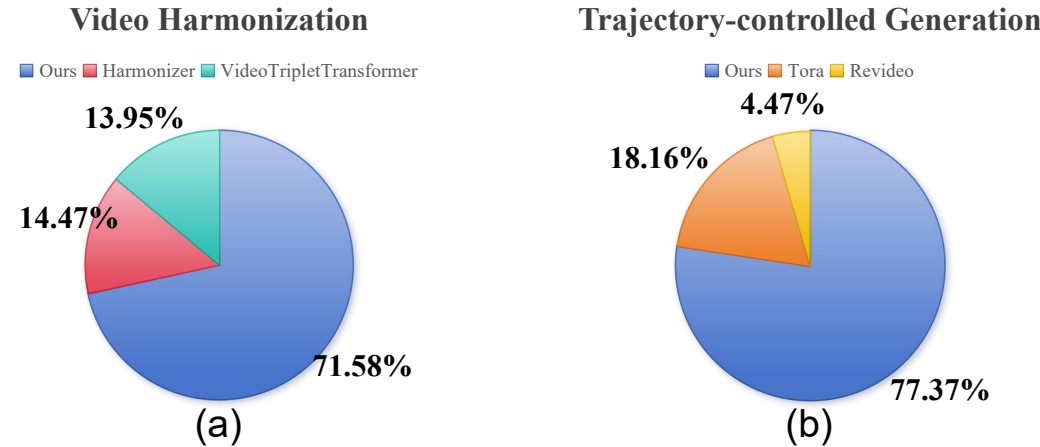

Figure 12: **User study comparison**. Our method receives the most user preference for both video harmonization(a) and trajectory-controlled generation(b).

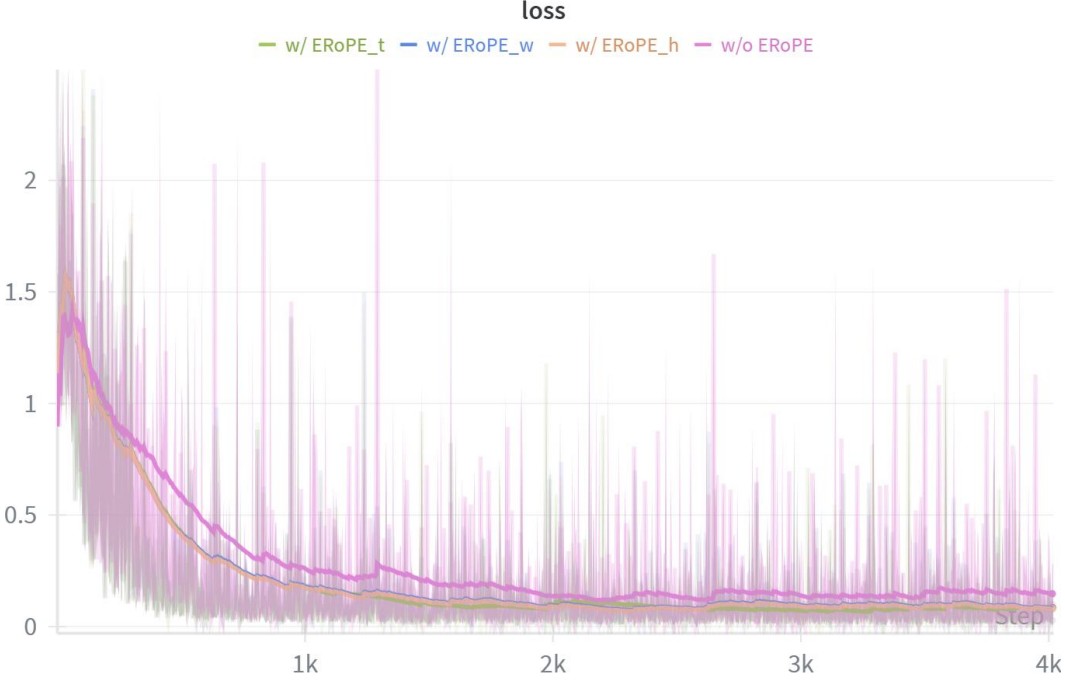

Figure 13: **Loss curves of applying ERoPE**. We apply ERoPE by extending along different optional dimensions, height, width, and timing. For comparison, we showcase the loss curve of ablation setting that applies the same RoPE on both masked video and foreground video, which is marked in pink. One can see that the three extension directions have equal positive impact on performance.

# E   USER STUDY

We conduct user study for both video harmonization and trajectory-controlled video generation, compared with Harmonizer and VideoTripletTransformer, Tora and Revideo, respectively. For each task, we provide 20 sets of visual comparisons and invite 19 professional volunteers to select their most preferred results. As shown in Fig. 12(a), most users prefer the video harmonization capability of our method, while Harmonizer and VideoTripletTransformer are equally favored. As to trajectory-controlled video generation, as shown in Fig. 12(b), our method obviously outperforms other related algorithms, which is consistent with the visual and quantitative results provided in the main paper.

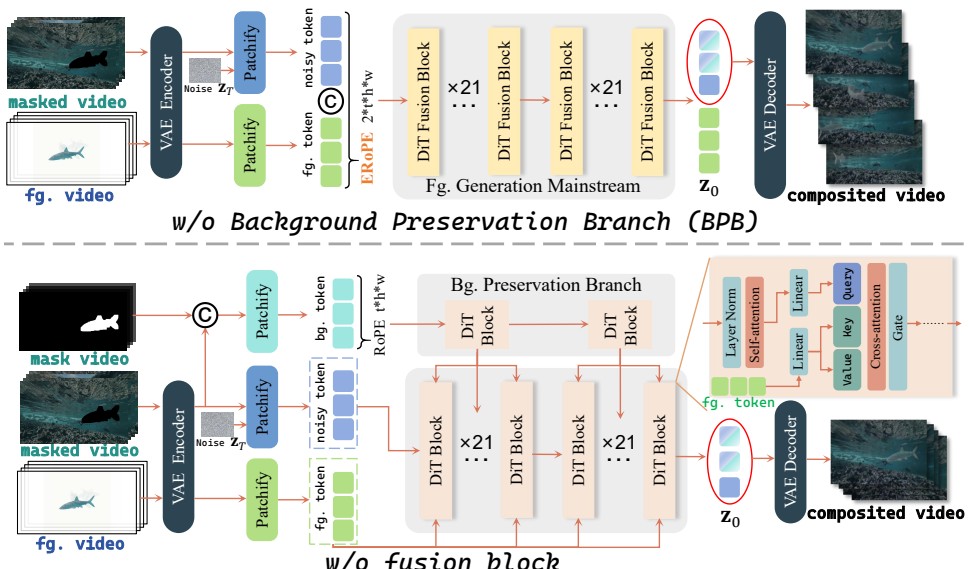

Figure 14: **Architectures of ablation settings**. We visualize architecture details of the two ablation settings, "w/o BPBranch" (upper) and "w/o fusion block" (lower). The former deletes the control branch, and the latter uses cross-attention to inject foreground condition instead of the proposed fusion block. Their control signal conversion is the same as full model. So we omit it and mainly showcase model designs.

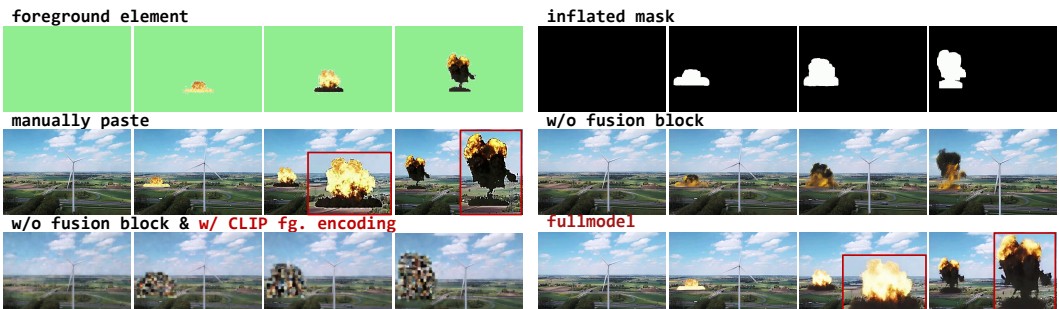

Figure 15: **Ablation on foreground CLIP encoding**. Based on the ablation setting of "w/o fusion block", we further replace the VAE encoding of the foreground dynamics with CLIP encoding (*i.e.*, "w/o fusion block & w/ CLIP fg. encoding") to validate whether semantic feature is more suitable for cross-attention. But results demonstrate that our model cannot utilize such an abstract feature.

# F  LOSS CURVE OF APPLYING ERoPE

To study the impact of extended position embedding in different dimensions on performance, we visualize the training loss curves of four settings in Fig. 13, where we apply ERoPE by applying position embedding along three dimensions, *i.e.*, height, width, and timing, which are marked as brown curve (w/ ERoPE_h), blue curve (w/ ERoPE_w), and green curve (w/ ERoPE_t), respectively. For comparison, we also attach the loss curve of an ablation setting as pink curve in Fig. 13, which is the training loss of using the same RoPE for both masked and foreground videos (w/o ERoPE). One can see that the training loss curves of three settings that uses ERoPE are all obviously lower than w/o ERoPE. Meanwhile, the loss curves of the three settings are highly consistent with each other, *i.e.*, the brown, blue, and green curves almost overlap. This comparison strongly prove the importance of our proposed ERoPE for utilizing layout-unaligned video conditions, and demonstrate the equality of these three extension directions. In the main paper, we use "w/ ERoPE_h". More importantly, we believe this discovery inspires future work for utilizing layout-unaligned video conditions.

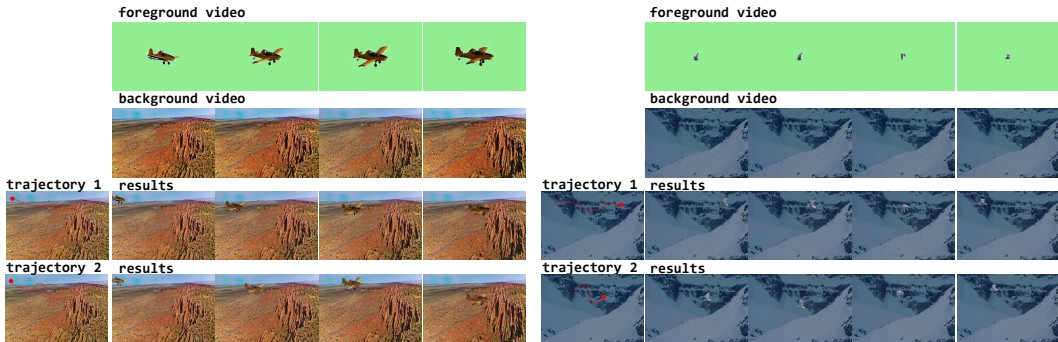

Figure 16: **Generating with different user-provided trajectories**. We apply two different trajectories to the same foreground-background video pairs. We can see that, given the same foreground and background videos with different trajectories, GenCompositor could generate different contents that strictly follow their corresponding trajectories.

## G  ABLATION ARCHITECTURES

As shown in Fig. 14, we visualize the architecture details of our two ablation settings, "w/o BPBranch" abd "w/o fusion block", mentioned in the main paper. The former eliminates the background preservation branch, but still concatenates foreground condition with noisy tokens at the spatial-level and utilizes the proposed DiT fusion block to fusing them, combined with the EROPE. Whilst "w/o fusion block" remains the background branch, but replaces the DiT fusion block with normal DiT modules. To inject foreground control, it utilizes cross-attention as shown in the lower of Fig. 14.

In addition, considering that previous cross-attention based methods mainly employ semantic features as external conditions and inject with cross-attention. For the ablation setting of "w/o fusion block" in the lower of Fig. 14, we also replace the VAE encoding for the foreground videos with CLIP encoding to validate whether semantic feature is more suitable for cross-attention than low-level feature of VAE. The corresponding visual results are given in Fig. 15, where we attach the foreground element and inflated mask in the first row. The second row shows a manually paste version as baseline reference, and results of "w/o fusion block" as ablation baseline. In the setting of "w/o fusion block & w/ CLIP fg. encoding", we use CLIP to encode foreground videos instead of VAE used in "w/o fusion block". One can see that the main content of background is still preserved, but the foreground conditions cannot be injected, and even the abstract semantic information that could have been inherited by "w/o fusion block" cannot be preserved, directly generating noise. We believe this is because using such a high-level abstract information, such as semantics, to generate low-level and dense visual signals (*i.e.*, video) is difficult. For example, using VAE features of foreground video can directly inject low-level messages of this dynamic condition, such as its shape and texture, and naturally only affects part region in generated video. But using CLIP features is theoretically equal to input an abstract description text, which directly impacts entire video, increasing learning difficulty. Moreover, fusing the abstract features (foreground video) extracted by CLIP with the low-level features (background video) extracted by VAE in a diffusion model is also challenging.

## H  INTERACTIVITY

In order to validate the flexible interoperability of the proposed method, we provide additional examples where the user specifies different trajectories and scale factors, respectively. In Fig. 16, given the foreground and background videos, we manually specify two different trajectories to GenCompositor for generation. The added elements exhibit different trajectories, which are highly consistent with the input trajectories. In Fig. 17, we provide three different scale factors in the same example. Obviously, the user-specified factors directly affect our mask videos and control the size of elements in the final results.

## I  MORE VISUAL RESULTS

We provide more visual results in Fig. 18 to showcase our superior video compositing capability. GenCompositor enables seamless video elements injection according to the user-given interaction. We visualize the designated trajectories as right dots and lines in the first frame of background videos. One can see that GenCompositor precisely injects foreground elements into the user-given positions and harmonizes their color. On the other hand, the background environmental changes caused by the inserted objects (such as the shadows in

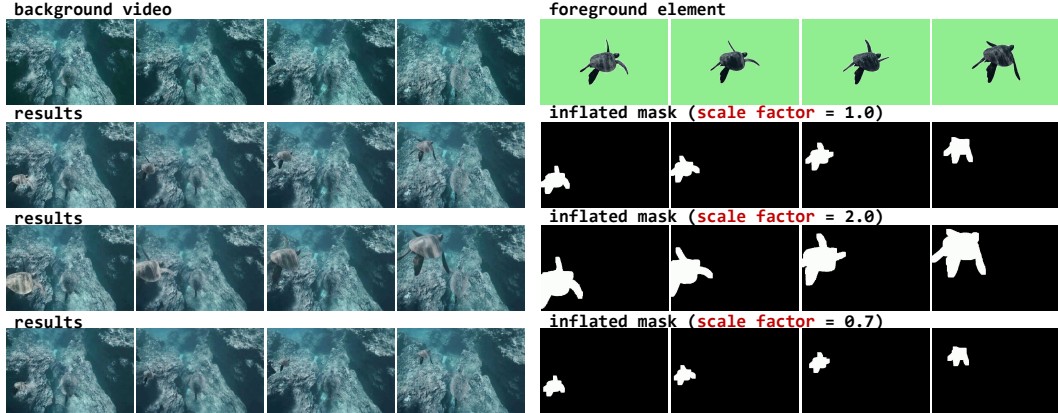

Figure 17: **Generating with different user-provided scale factors**. Our method produces mask videos that follow user-provided factors and control the size of added elements in final results.

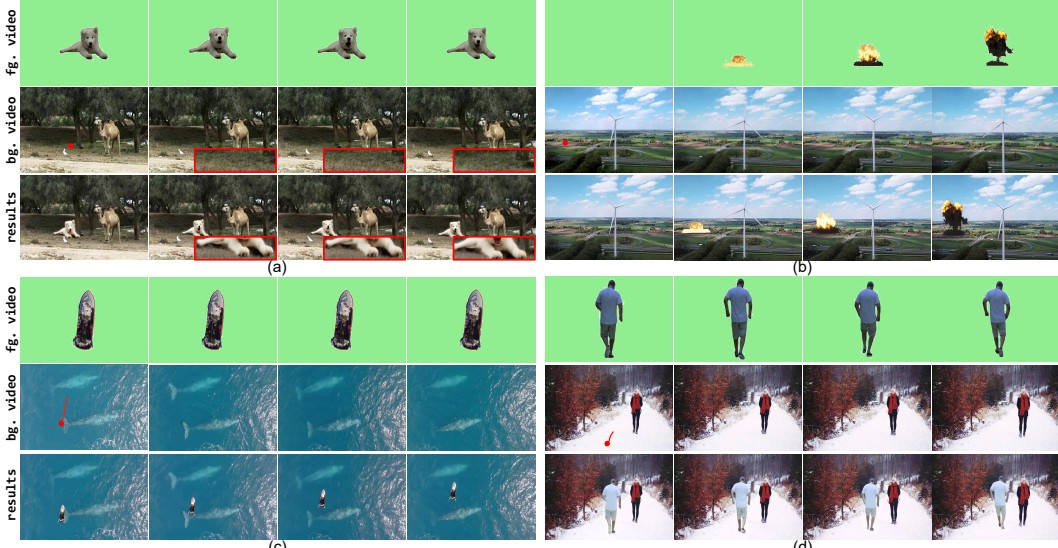

Figure 18: **More visual results of GenCompositor**. Our method enables seamless video elements injection and user interaction. On the one hand, it precisely injects foreground elements into the user-given position and harmonizes their color. On the other hand, the background environmental changes caused by the inserted objects (such as the shadows in the red box) are also automatically predicted by our generative model, proving the superiority of generative video compositing.

the red box) are also automatically predicted by our generative model, proving the superiority of generative video compositing task.

## J  DISCUSSION

As the first attempt in generative video compositing, this paper proposed a new practical video editing task, provided the first feasible solution, and curated the first usable training dataset. Interestingly, we found that the proposed new task, generative video compositing, naturally supports other related tasks such as video inpainting and video element removal. More importantly, for video compositing itself, the proposed GenCompositor allows users to specify arbitrary trajectories and object sizes changing with timing. It means that users even simulate 3D spatial effects, such as near-large-and-far-small. We provide some examples in supplementary video to demonstrate this function. In addition, some techniques in this paper are novel and have the potential to be applied to other similar application scenarios. For example, the proposed ERoPE can fully utilize layout-unaligned video conditions, and experiments in this paper demonstrate that, for video signals, the optional three extension directions (*e.g.*, width, height, and timing) are equal, as said in Sect. F. We believe this contribution also plays a significant role in other video editing tasks.

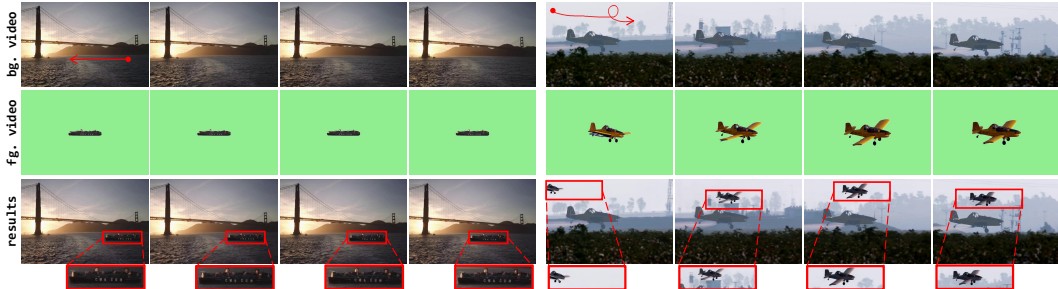

Figure 19: **Our performance in extreme background lightness**. For the left, the freighter itself is not illuminated. The composited results shows a freighter illuminated from left to right, with a warm, sunset-like glow, which is consistent with background content. For the right, the added aircraft itself has a high exposure intensity but background is unexposed. In output, our model adaptively adjusts lightness of added element to make the entire result coordinated.

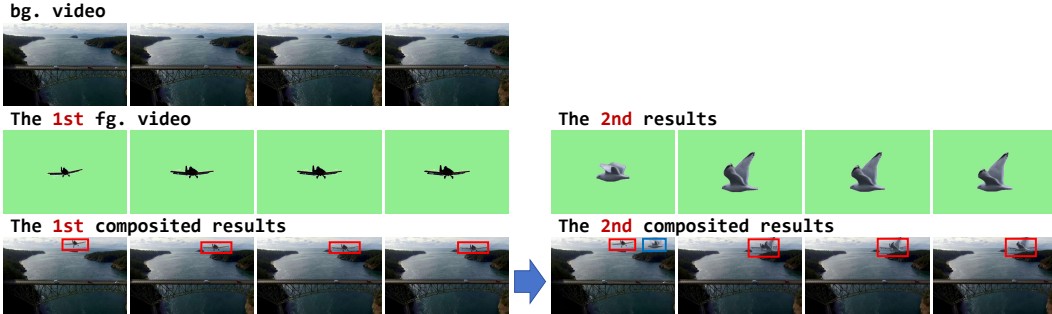

Figure 20: **Our performance on multiple objects compositing**. We showcase the background video to be edited in the first row, and two element videos in the second row. As shown in the third row, on the left, we use red box to highlight the inserted airplane. On the right, we additionally add a seagull; the airplane and the seagull are partially obscured.

**Future work.** Although this paper enables video injection and realistic interaction, some challenging topics still exist in extreme conditions. For example, we employ gamma correction, a simple yet effective augmentation, to address most common cases well. As shown in Fig. 19, GenCompositor has been able to generate harmonious results by adaptively adjusting lighting and filter effects of foreground elements to match challenging background brightness. But it may not be robust enough for sophisticated extreme background lighting. Meanwhile, the added elements cannot undergo complex occlusion changes with environment. But as the first attempt, GenCompositor actually has been able to generate videos with impressive occlusion effects. As shown in Fig. 20, we add an airplane and a seagull to a single background video, and GenCompositor tackles this case well. We believe more extreme background lighting can be solved by replacing other luminance augmentation methods, and the more challenging occlusion can be achieved by adding depth-aware or 3D priors. Since GenCompositor already works for most scenarios, we leave these issues for future work. We believe a potential research direction in generative video compositing is enabling novel-view foreground prediction and new-oriented compositing, which may be addressed by introducing 3D priors and we leave it in the future work.

## K   INTERACTIVE WEB APP

To ease utilization of the proposed GenCompositor, we realize an interactive web demo in our code, which can be found in supplementary materials. Specifically, our interactive demo is shown in Fig. 21. We firstly list the essential operating tips, and provide some available video samples in the first box. Users can directly specify background and foreground videos from them, or choose not to use them and upload their own materials. In step 1, users segment foreground element from the foreground video. In step 2, users specify the movement trajectory in the background video. Finally, in step 3, users can specify varying rescale parameters, which is defaulted as 0.4, to assign the scales of added element, and compositing final video through our model. Notice that we also provide an extended application in the box in the lower right corner, which can inpaint video or remove dynamic elements from the video. In this function, users do not need to specify foreground video. Our code automatically selects the blank video as foreground, and removes the specified element from the video smoothly.

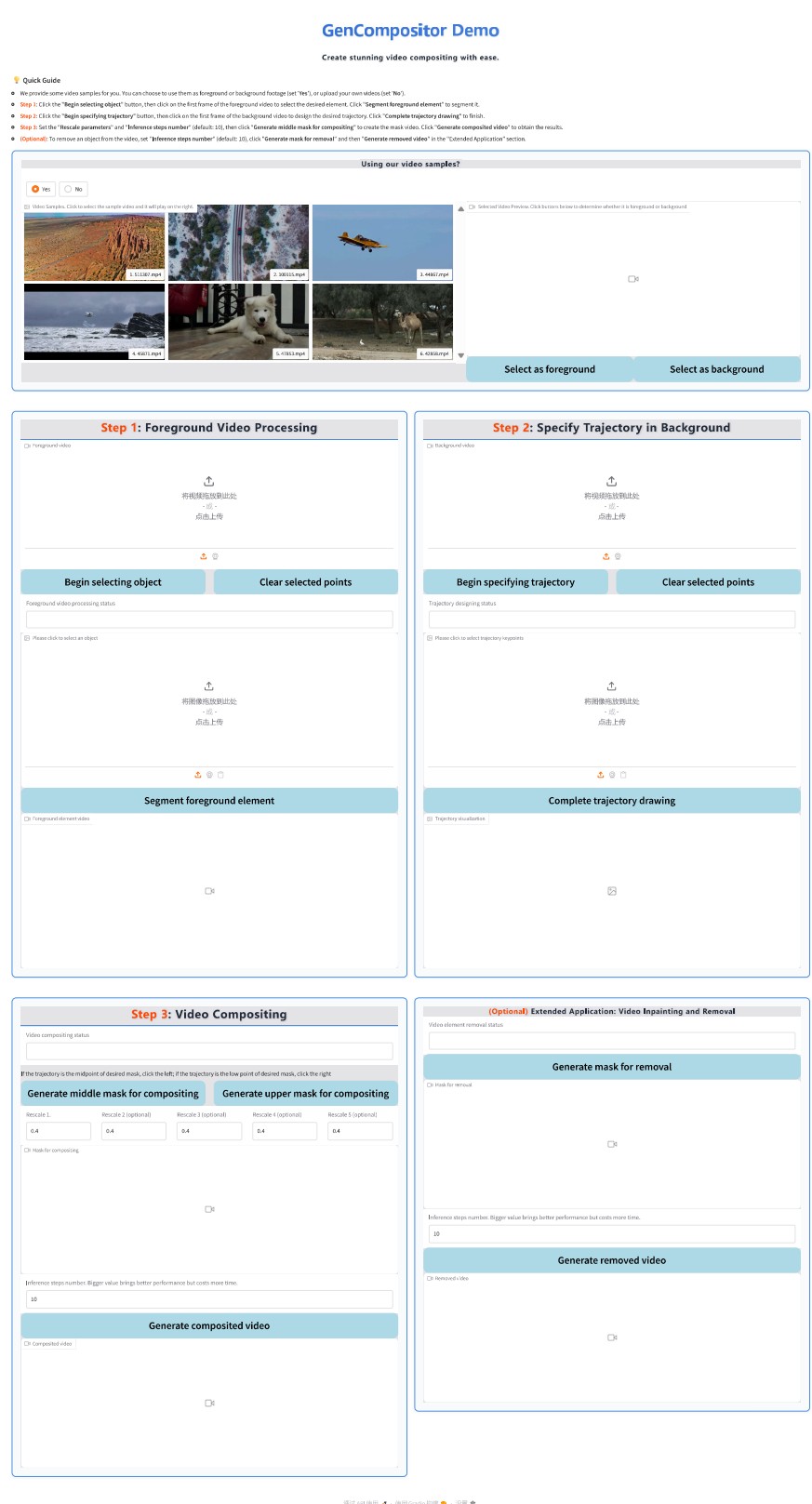

Figure 21: **Interactive interface of our web app**. As an interactive video editing method, we realize an interactive web demo to ease the utilization of GenCompositor. Related code can be found in supplementary materials.

