# OpenReview forum: "GenCompositor: Generative Video Compositing with Diffusion Transformer"
_ICLR.cc/2026/Conference — ICLR 2026 Poster_

### Official Review · Reviewer_7r6X · 2025-10-16

**Soundness:** 3
**Presentation:** 3
**Contribution:** 3
**Rating:** 6
**Confidence:** 4

**Summary:**

This paper proposes GenCompositor, a generative video compositing method that can adaptively inject foreground video with user control into the background video. The proposed framework was developed based on the diffusion transformer pipeline, allowing users to control the size, trajectory of the foreground objects, and seamlessly preserve them in the background video. To validate the effectiveness of this proposed framework, the paper also curated a dataset called VideoComp. The experiments have been validated to compare against both trajectory-controlled video generation and non-trajectory-controlled video generation methods.

**Strengths:**

Overall, this paper proposes a new task, i.e., generative video compositing, which has not been well addressed before. This differentiates from existing works on video compositing as it utilizes the generative models to complete the task. From this perspective, the proposed task is similar to the existing one, but tackles the problem from a different technical perspective.

In addition to the task, the paper shows some strong points in curating the dataset required to evaluate on this task. If released, this should help further advance the research in this field. The paper offers a good motivation for why this problem is important and presents the components in the pipeline relatively well. Below, I will show the weak points that need particular attention.

**Weaknesses:**

The experimental visualization was not quite clear. For instance, by checking on Figure 1, what is the foreground object on the left top example of the figure? The depicted trajectory is also unclear in this case. It will be nice to have high-resolution and good-quality results. Similarly in Figure 7, the trajectory is not very clearly depicted.

The paper mentions the interaction between the added foreground object and the background, citing the explosion effect in Figure 1(a) as an example. This looks very interesting. However, I don't believe this paper presents sufficient technical details to explain why that happens. Usually, we will expect the added element to have a static presence. But this claim shows that the newly added element can adapt to the background content. This part is not supported by the technical details, at least not clearly presented. This is related to mask inflation, but not entirely.

Regarding the input conversion, the user trajectory has been used to generate the mask video and masked video. The trajectory itself was not directly encoded. It will be nice to directly state what the conditions of "c" are in Eqn. (2).

Based on the workflow of GenCompositor in Figure 2, the masked video and its corresponding mask video has been concatenated as the input. This indicates that the GenCompositor deals with the layout-aligned conditions. In Section 4.4, where ERoPE was discussed, the problem has been shifted to the layout-unaligned conditions. This creates some gaps in understanding, as the workflow in Figure 2 does not incorporate the layout-unaligned conditions. Will the layout-unaligned conditions change the workflow of the proposed GenCompositor? This creates some confusion here.

The role of the inflated mask could be better clarified. Initially, I was thinking that the mask regions were expanded to increase the size of the foreground object, if there is an imperfect foreground mask. A Gaussian filter was used to inflate the mask video. When looking at Figure 8, I have also noticed that the inflated mask was naturally generated as the foreground object changes.

In summary, the paper tackles an interesting task, and the presented pipeline is also feasible to address the proposed task. There are some technical contributions regarding the proposed pipeline, as well as the ERoPE part. The proposed dataset is also a good contribution. Therefore, the paper looks positive. However, the paper also shows the concerns listed above, which need to be addressed well. The corresponding answers to address these concerns would be critical for the final decision.

**Questions:**

1. The interaction between the added foreground object and the background is unclear. How this is related to the technical component in the proposed pipeline needs to be justified.

2. The input conversion procedure is unclear and needs to be further explained.

3. The relationship between layout-unaligned conditions and its proposed pipeline is not clearly justified.

4. The role of the inflated mask and its generation process needs to be better explained.

---

> ### Author Response · Authors · 2025-11-23
>
> **1. The experimental visualization was not quite clear. (Fig.1 and Fig.7)**
>
> For Fig.7, following your advice, we have carefully thickened trajectory curves in this revision to highlight the trajectory. For Fig.1, as mentioned in Lines 81-87, the foreground object in the top left example is the flames of the explosion. As mentioned in Lines 201-203, when users only click a point (as shown in Fig.1(a)), our method will automatically track the trajectory of this point in subsequent frames based on the optical flow. We added a specific introduction in Lines 84-85 in this revision to emphasize the trajectory in Fig. (a), which is marked in RED.
>
> **2. The interaction between the added foreground object and the background.**
>
> In our task of generative video compositing, the added object and its surrounding background are jointly generated by the diffusion model within the inflated mask (Sec. 4.1, 4.2). Concretely, the condition of our model includes (i) background video features, (ii) the inflated mask video, and (iii) the masked video; all are fused through shared self-attention in the DiT backbone. This design forces the model, during training, to generate pixels in the region around the object using both foreground and background cues. As a result, when we place an object near an explosion (Fig. 1(a)), its shading, transparency, and even its surrounding pixels adapt to the background content, rather than remaining a static overlay. According to your professional advice, we made this mechanism explicit in Sec.4.1 of this revision; the revised part is marked in RED.
>
> **3. The input conversion procedure.**
>
> We thank the reviewer for pointing out this ambiguity. In the revised manuscript, we explicitly define how the user control and trajectories enter Eq. (2). We carefully revised the first paragraph of Sec.4.1 according to your advice and marked it in RED.
>
> Concretely, in Sec. 4.1 we now let the user control be u={γ,s}, where γ is the 2D trajectory on the background video and s is the rescale factor. We deterministically rasterize u into a binary mask video M and thereby get a masked background video X = (1−Mt)⊙vb, where vb is the input background video. We then set the conditioning variable in Eq. (2) to c=(M,X), i.e. the network is conditioned on the mask video and masked background video derived from the user trajectory and scale, while the foreground video vf provides identity and motion cues. The updated text makes clear that the raw trajectory is not directly encoded as a separate input; instead, it is fully represented by these rasterized videos that supervise where and how the foreground element should be composited.
>
> **4. The relationship between layout-unaligned conditions and the proposed pipeline.**
>
> We sincerely appreciate the reviewer for pointing out this confusion. In this version, we carefully clarify this issue at the beginning of Sec. 4.4, and marked the revision in RED.
> Specifically, we actually distinguish layout-aligned and layout-unaligned conditions with respect to the target layout. In Fig. 2, the mask video M and masked background video X are layout-aligned with the background and target video (vb, z0), while the foreground video vf is layout-unaligned (its dynamic content is centered and follows a different camera frame). Sec. 4.4 is then rewritten to state that ERoPE is applied inside the DiT to jointly embed (vb ,M,X,vf ): RoPE is kept for layout-aligned videos, and ERoPE assigns separate positional labels to layout-unaligned videos, without changing the data conversion or compositing workflow shown in Fig. 2. We believe this makes the relationship between layout-unaligned conditions and the proposed pipeline much clearer.
>
> **5. The role of the inflated mask and its generation process.**
> We deeply appreciate the reviewer’s request for a clearer explanation of mask inflation. In the revised Sec. 4.1, we make its role and generation process explicit.
>
> In Sec. 4.1, we now distinguish the binary foreground mask Mbin from the inflated mask M. Given Mtbin for each frame, we first blur it with a Gaussian kernel and then threshold it to obtain an expanded “editable buffer band” around the object. This inflated mask is used both to build the masked video and as the mask condition fed to the network. Conceptually, mask inflation does not aim to artificially enlarge the foreground object; instead, it defines a slightly larger region where the model is allowed to modify the background for better harmony and small mask inaccuracies are tolerated. We also clarify in the ablation paragraph (Fig. 8) that the first row “inflated mask” is exactly this Gaussian-inflated mask, deterministically computed per-frame from the foreground segmentation.

---

> ### Author Response · Authors · 2025-11-27
>
> Dear reviewer, thank you very much for your constructive comments and professional advices. According to your advices, we have carefully revised the manuscript, and we sincerely believe that your feedback has greatly helped improve the quality of our final paper. If your have any further concerns, please let us know and we will try our best to address any remaining concerns. Thank you again for your time, effort, and professional advice on our manuscript!

---

### Official Review · Reviewer_irsQ · 2025-10-25

**Soundness:** 3
**Presentation:** 3
**Contribution:** 2
**Rating:** 4
**Confidence:** 4

**Summary:**

The paper introduces GenCompositor, a diffusion-transformer-based framework for generative video compositing, enabling insertion, removal, and harmonization of dynamic foreground objects within videos. The method takes as input a masked background video, a mask video, and a foreground video, along with user-defined trajectory and scale controls. The model architecture follows a  MMDiT, similar to those used in FLUX, SD3, or HunyuanVideo, with full self-attention, and includes:
- A background preservation branch to maintain scene consistency,
- A DiT fusion block, which concatenates background and foreground tokens for fusion instead of using cross-attention, and
- An Extended Rotary Position Embedding (ERoPE) to mitigate layout misalignment artifacts.

Furthermore, a new dataset, VideoComp, containing 61K video triplets, is curated for training. The qualitative results are visually strong and demonstrate convincing object insertion and harmonization, though the quantitative metrics show only modest improvements over existing methods.

**Strengths:**

- **Strong engineering effort:** The system is well-implemented and technically solid. The architecture is clean and the pipeline is carefully designed.
- **Good visual quality:** Qualitative and video results are impressive, showing smooth motion and coherent integration of foreground and background.
- **Clear writing and presentation:** The paper is well-organized, easy to follow, and supported by clear figures and a detailed video presentation.
- **Practical relevance:** The task aligns well with real-world video editing workflows and could be useful for production or creative pipelines.

**Weaknesses:**

1. **Limited conceptual novelty**:
    - The core components are incremental modifications of existing DiT architectures.
    - The background preservation branch resembles ControlNet-style conditioning, which is also stated in the paper.
    - ERoPE is a simple positional embedding shift; while useful, it is not conceptually innovative.
Overall, the work feels like a strong system-level integration rather than a conceptual contribution.

2. **Inadequate quantitative evaluation**:
    - The paper reports only frame-level metrics (PSNR, SSIM, LPIPS, CLIP). These are not sufficient for video generation; video-level metrics such as FVD or KVD are missing.
    - Quantitative improvements are relatively small, making it difficult to assess significance.

3. **Weak baseline comparisons**:
    - The compared baselines (Harmonizer, VideoTripletTransformer) are both video harmonization methods, not true generative compositing or conditional video generation systems.
    - DynVFX can be used for comparisons. Although it cannot add directly the reference objects, it can add objects through text-prompts, which will also strengthen the paper's results.

4. **Dataset concerns**:
    - The construction of the new VideoComp dataset is clearly described, combining 409K source videos into 61K compositing triplets via a semi-automatic pipeline (SAM2 + CogVLM + Qwen). While this is a solid contribution, the paper provides limited analysis of dataset characteristics such as category diversity, motion distribution, or domain coverage, which limits understanding of how well the dataset covers diverse motion and scene types. Including a brief quantitative summary would strengthen this part.

5. **Conceptual framing and scope**:
    - The paper frames generative video compositing as a new task, but it heavily overlaps with existing controllable video generation and video inpainting setups.

**Questions:**

- The results without the fusion block are much worse. Could this be due to the limited representational power of the VAE encoder? I would like to see additional results where a stronger feature extractor, such as DINO or CLIP, is used for the cross-attention variant, as this could provide more meaningful semantic conditioning.
- The model depends strongly on mask and trajectory inputs. How flexible is it when these are imprecise or unavailable? Can it generalize to more unconstrained compositional setups?
- How well does it handle multiple objects or occlusion?
- The paper mentions luminance augmentation for harmonization. Is it sufficient for handling complex illumination changes, or does it fail in extreme lighting conditions?

---

> ### Author Response · Authors · 2025-11-23
> **Response to Weaknesses**
>
> **1. Limited novelty**
>
> We fully understand the reviewer’s concerns. (1) While the individual modules may not be conceptually novel on their own, we are the first to combine and adapt these technologies for this new task, establishing the first feasible pipeline for video compositing. Our paper includes comprehensive ablation studies to demonstrate why these specific techniques are effective, as opposed to existing alternatives. (2) Additionally, we introduce important advancements, such as ERoPE, which addresses a fundamental limitation of RoPE for unaligned video conditions. This approach benefits not only video compositing but also broader layout-unaligned video conditioned generation tasks. (3) Importantly, as a pioneering effort, we explored various possible solutions to address this challenge. We thoroughly analyze these unsuccessful attempts within the paper, providing valuable insights for future research. (4) Finally, we commit to releasing the first large-scale dataset for this task (VideoComp), representing a significant practical contribution to the community.
>
> **2. Inadequate quantitative evaluation**
>
> Following your advice, we report FVD and KVD results in Tab. 2 of revised version. We also include VideoBench, a widely used metric for measuring feature distribution differences between real and generated videos, similar to FVD.
>
> Since the numerical values of the metrics are not very sensitive to the results, the quantitative improvements are relatively small. And we find that the improvements on VideoBench of our methods are large compared with the improvements of previous methods. For example, Tab.2 in FramePack (NeurIPS 2025) shows that the best method has advantages of 0.02% (Aesthetic), 0.51% (Motion), and 1.2% (Identity) over the 2nd-best method. Correspondingly, in our Table 2, our method has advantages of 2.67% (Aesthetic), 0.66% (Motion), and 1.31% (Identity).
>
> Moreover, as a video editing task, metrics on generation quality are used to verify the output's quality rather than editing performance. The key evidence for the effectiveness of our method comes from the user study in Sec. E and qualitative results, which demonstrate that our approach reliably respects layout, motion control, and foreground dynamics consistency.
>
> Video harmonization is typically regarded as a low-level vision task, which uses pixel-level metrics (e.g., PSNR, etc.) instead of feature-level (e.g., FVD, etc.) metrics to measure quality, so here we follow the classic practices in this field.
>
> **3. Weak baseline comparisons**
>
> Since the code for several recent image-conditioned video object insertion methods (e.g., VideoAnydoor, **DynVFX**, UNIC) has not yet been released, we compare our approach with the most recent open-source method, VACE, which supports inserting object ID from a given image, to strengthen our results. Corresponding results have been highlighted in red and updated in Tab.2 and Fig.7.
>
> It is important to note that existing object insertion tasks differ from our setting: they generally focus on inserting object ID from a static image and generating realistic dynamics. In contrast, our task is designed to faithfully inherit both object ID and the original foreground dynamics.
>
> **4. Dataset concerns**
>
> According to your suggestions, we add three pie charts to analyze category diversity, motion distribution, and domain coverage of VideoComp. We found that this analysis strongly supports the validity of the content in the proposed VideoComp dataset. This part has been added in Fig.10 and Sec.B.3 in the supplementary material. Please refer to the revised part in our newly submitted manuscript.
>
> **5. Conceptual scope**
>
> In Lines 67-77 of this paper, we discuss similarities with existing tasks, but also explain that the solutions of these tasks cannot address the real requirements of video compositing.
>
> Controllable generation (e.g., ID insertion) receives text or image, neither of which can use a full dynamic video as a condition. This difference leads to different objectives in fine-grained motion inheritance and appearance dynamics. The temporal conditions cannot be obtained from image or text, which has been confirmed in this paper: even with the same trajectory, Tora and Revideo fail to preserve the dynamic shape and motion of the foreground element.
>
> Video inpainting only aims to fill holes. Our task injects externally-sourced dynamic content, which requires: cross-video temporal alignment, appearance harmonization, and compositional blending. These capabilities do not exist in the inpainting formulation.

---

> ### Author Response · Authors · 2025-11-23
> **Response to Questions**
>
> **Q1: Using semantic feature extractor for foreground conditions**
>
> According to your suggestions, we add an experiment that uses the CLIP encoder to encode the foreground video used for the cross-attention to replace the VAE encoding of the foreground video. Intuitively, CLIP extracts high-level semantic features, which means motion details and appearance are more difficult to retain by CLIP, which is not conducive to video compositing, according to the experiment. We attach visual experiments in Fig.15, and the corresponding discussion (marked in RED) in Sec. G of the supplementary materials, which strongly validates the limitations of replacing VAE with this feature encoder.
>
> **Q2: Imprecise mask and trajectory inputs**
>
> In fact, our model is robust to inaccurate mask and track as described in Sec. 4.1. Where we apply mask inflation to create a buffer region around the intended object area. This is explicitly designed to absorb pixel-level misalignments, make the model robust to imperfect SAM2 boundaries, as evidenced by the “w/o mask inflation” vs. “full model” comparison in Fig. 8 and Table 3.
>
> Meanwhile, optical flow algorithms are not necessary for the system. Normally, users can drag a trajectory instead of clicking and using optical flow to track. It is only required in the case of “click-only” as another option of dragging a trajectory.
>
> More importantly, we also create training data based on the same preprocess tools as said in Sec.B. This means during training, our model also receives imprefect input and tries to generate results similar to Ground Truth video. This end-to-end framework, combined with natural denoising ability of video diffusion model, gives our final model a certain degree of correction capability.
>
> **Q3: Multiple objects and occlusion**
>
> We deeply appreciate your constructive suggestion. According to this advice, we supplement discussions about occlusion effects in Sec.J, where we have discussed that a challenging topic remaining in this paper is compositing multiple elements or addressing occlusion effects.
>
> But as the first attempt, GenCompositor has actually been able to predict impressive occlusion movements. Related results are shown in Fig.20 of this revision. Please refer to our revised manuscript. We add two elements, an airplane and a seagull, which are partially obscured by each other. GenCompositor realistically predicted this effect. This strongly demonstrates our model, as the first attempt, has been able to tackle such a challenging effect. But we still point out that introducing 3D or depth priors will benefit this issue.
>
> **Q4: Extreme lighting conditions**
>
> We supplement discussions about extreme lightness variation in Sec.J, where we have discussed that a challenging topic remaining in this paper is extreme background lightness. But as the first attempt, we have proposed a simple yet efficient solution that has been able to tackle most common situations.
>
> We provided some demos with rare background lightness in Fig.19 of this revision. One is the setting sun, casting light from a direction. The other is a gray sky background, giving a cool-toned filter effect. GenCompositor adaptively adjusts the corresponding lightness in added elements to generate visually harmonious results. Related discussion is added in Sec.J and marked in RED. We believe this revised version strongly supports our effects in challenging lightness conditions.

---

### Official Review · Reviewer_ThTu · 2025-10-28

**Soundness:** 3
**Presentation:** 3
**Contribution:** 3
**Rating:** 4
**Confidence:** 4

**Summary:**

GenCompositor introduces generative video compositing, an automated task to inject dynamic foreground elements into background videos via user-specified trajectories/scales. Its core is a Diffusion Transformer pipeline with three key components: a lightweight background preservation branch ensuring input-output consistency, DiT fusion blocks with full self-attention for foreground-background token fusion, and Extended Rotary Position Embedding (ERoPE) addressing layout misalignment. Trained on the 61K-video VideoComp dataset, it outperforms video harmonization (e.g., Harmonizer) and trajectory-controlled generation (e.g., Tora) methods in PSNR, SSIM, and motion smoothness. Ablation studies validate the necessity of each component. GenCompositor enables seamless, interactive video editing with realistic foreground-background interactions.

**Strengths:**

1. The work addresses an underexplored yet challenging setting: interactive, adaptive injection of foreground identity and motion into target videos, enabling fine-grained control over size, trajectories, and other dynamic attributes.

2. The showcased results demonstrate strong perceptual quality and temporal coherence, suggesting the method’s practical utility.

3. The release of implementation details and code substantially improves reproducibility and facilitates future research.

4. The construction of the 61K-pair VideoComp dataset represents a significant data-engineering contribution and a useful resource for benchmarking video composition.

**Weaknesses:**

1. In my opinion, the innovation of the module is more of an engineering improvement rather than a fundamental breakthrough. Specifically:
- For Sec.4.1, the proposed Input Conversion primarily involves operations like scaling and expanding the mask foreground object, which are quite common in existing methods.
- For Sec.4.2, the use of a black area to indicate the modified background and a binarized mask video to represent the masked area is also a common practice in prior work.

2. Although the RoPE alignment appears to be a promising idea for alignment, the authors did not provide sufficient elaboration on this design.

3. Additionally, it is suggested to revise Fig.3(A) to align with the paper's description by using a mask video instead of a foreground video, better demonstrating the unique application of ERoPE.

**Questions:**

1. Sec.4.3 proposes a new token-wise concatenation method. However, what differentiates it from in-context learning approaches like VACE [Jiang2025 et al.]? Perhaps I missed some details, but I did not find a detailed explanation in the ablation study or supplementary materials.

2. After reviewing the supplementary material on your data construction, I have a question: how do you ensure that the foreground data strictly adheres to the user-provided trajectories, especially when considering variations in angle and direction? This raises concerns about the strictness of the task’s generalizability.

---

> ### Author Response · Authors · 2025-11-23
>
> **1. Technical contribution**
>
> We understand that the preprocessing in Sec. 4.1 and 4.2 are not conceptually innovative. However, we are the first to combine these technologies for this new task of generative video compositing. This paper conducts sufficient ablation to prove why these operations work in this task. Moreover, the operations in Sec. 4.1 and 4.2 are only the preprocessing of our methods. The main technical contributions of our work lie in (1) token-wise concatenation, (2) DiT Fusion Block, and (3) ERoPE, which are listed in Sec. 4.3 and 4.4. We sincerely hope you consider these real technical contributions.
>
> **2. Explanation on ERoPE**
>
> We sincerely thank the reviewer for highlighting that the ERoPE design was insufficiently explained. In response, we have added a detailed elaboration of this design in Sec. 4.4 of the revised manuscript, with the new content marked in red.
>
> **3. Revise Fig.3(A)**
>
> We boldly guess you are referring to modifying Fig.4(a). In Fig.4, ERoPE is used to assign indices for two misaligned videos: the masked video to be edited and the foreground video serving as the dynamic condition. Notably, the mask video is **not** processed by ERoPE. Because the mask video is layout-aligned with the background video, it utilizes standard RoPE to preserve the background content, as illustrated in Fig. 2.
>
> **Q1: Difference with VACE**
>
> The in-context learning in VACE and token-wise concat in GenCompositor are similar, but the implementations differ. VACE concatenates an image along the temporal dimension of the video and then patchifies the combined sequence once, treating it as a “context frame” that provides semantic-level guidance. In contrast, our method first patchifies the background and foreground videos separately, then concatenates them at the token level using disjoint ERoPE. This approach prevents feature interference between the foreground and background videos during tokenization, explicitly separating these two layout-unaligned videos.
>
> **Q2: Adhere to the user-provided trajectories**
>
> Different from existing ID injection tasks, which generate new motion with different angles and directions according to the trajectories, the setting of classical video compositing task does not modify the angles of foreground elements and aims to faithfully inherit their motion. Our work focuses on automating this process under the same task setting as the first generative approach. Specifically, if users wish to composite a bird flying to the left, they should select or pre-capture a foreground video showing a bird already flying in that direction as the footage.

---

> ### Author Response · Authors · 2025-11-27
>
> Dear reviewers, we sincerely appreciate for your professional comments on our manuscript. We have carefully revised our paper and responded to your comments one-by-one. Your valuable feedback has significantly improved the quality of our manuscript. We kindly invite you to provide further feedback if you have any concerns, and we are ready to do our best to address any issues you may have. We would be extremely grateful if you could consider reasonably increasing final rating based on our responses and the revised paper.

---

> ### Comment · Reviewer_ThTu · 2025-11-28
>
> I remain cautious about the innovation of the paper; however, the authors' revisions and supplementary content have addressed my other concerns well. Taking into account the comments from other reviewers, I believe the score can be increased appropriately.
>
> Due to the large-scale systematic impact of this information leakage incident on the review process, we still need to wait for the handling decisions from the ICLR organizing committee. I will make a final decision based on the situation.

---

### Official Review · Reviewer_sd3W · 2025-10-31

**Soundness:** 3
**Presentation:** 3
**Contribution:** 2
**Rating:** 6
**Confidence:** 3

**Summary:**

This paper introduces a new video editing task called generative video compositing and proposes the first feasible solution, GenCompositor. Based on the Diffusion Transformer architecture, this method can automatically composite dynamic elements from a foreground video into a background video according to user-specified trajectories and dimensions.

**Strengths:**

1. This work is the first to introduce the "generative video compositing" task, which aims to automatically composite foreground video elements into a background video using generative models while supporting user control over attributes such as trajectory and scale.
2. A complete architecture is proposed, including a background preservation branch, DiT fusion blocks, and Extended Rotary Position Embedding (ERoPE), aiming to address the three main challenges: background consistency, foreground injection, and user control.

**Weaknesses:**

1. Appendix F shows that extending RoPE along height, width, or temporal dimensions yields nearly identical training loss curves, leading authors to conclude these three directions are equivalent. However, this dimension-agnostic behavior seems contradictory to the inherent spatial-temporal characteristics of videos. If ERoPE truly models spatial layout relationships, spatial extensions (height/width) should outperform temporal extension since they directly encode spatial proximity. Conversely, if it captures motion dynamics, temporal extension should be more effective given the causal and directional nature of video frames. The complete equivalence across all three dimensions suggests that ERoPE may simply function by expanding the position encoding to avoid feature conflicts, rather than genuinely modeling spatial-temporal interactions between foreground and background. Could the authors clarify this dimension-independence phenomenon and explain whether simpler alternatives (such as learnable position offsets for different video sources) might achieve comparable results?

2. The practical usage requires SAM2 for foreground segmentation and optical flow algorithms for trajectory tracking, and these preprocessing steps may introduce errors and increase system complexity, yet the paper does not discuss how these errors propagate and accumulate to affect the final results.

3. The paper only compares its method with two video harmonization approaches and two trajectory-controlled generation methods, but fails to include comparisons with recent state-of-the-art methods in related tasks such as video object insertion, which share similarities with generative video compositing.

**Questions:**

Please refer to the weaknesses.

---

> ### Author Response · Authors · 2025-11-23
>
> **1. Discussion on ERoPE**
>
> We thank the reviewer for this insightful comment. Although the semantics of HWT dimensions in video are different, all of these video features will be flattened into one dimension after tokenization. Consequently, the network processes these tokens with no distinction between the three dimensions. From a mathematical perspective, we have added more discussion about the proposed ERoPE in Sec.4.4 of this revision, which is marked in RED. As clarified in Eq. (8), ERoPE introduces a shift in the frequency phase of tokens. Applying it along height, width, or time is mathematically equal to moving video as a rigid body in the phase space (i.e., Δs and Δs' in Eq.9). In practice, we find that this ``separation`` is more important than the specific dimension along which the shift is applied.
>
> Regarding the suggested learnable position offsets, we agree that this is a natural extension. However, this introduces additional training parameters into the network, which is more computationally expensive and such an implicit manner may not be effective during training. In contrast, ERoPE is a parameter-free mechanism designed to disambiguate layout-unaligned videos without introducing extra computational costs.
>
> **2. Imperfect mask and optical**
>
> We understand the reviewer’s concern regarding the error propagation of SAM2 for segmentation and optical flow for trajectory tracking, and have added related discussion in Lines215-248 of this revision, which is marked in RED. The errors may occur when the SAM2 or optical flow algorithms totally fail. On the other hand, our system has a certain degree of robustness to these errors.
>
> ``(1)`` As described in Sec. 4.1, we apply mask inflation to create a buffer region around the target object. This buffer is explicitly designed to absorb pixel-level misalignments, enhancing the model’s robustness to imperfect SAM2 boundaries, as evidenced by the “w/o mask inflation” vs. “full model” comparison in Fig. 8 and Tab.3. ``(2)`` Meanwhile, optical flow algorithms are not necessary for the system. Normally, users can drag a trajectory instead of clicking and using optical flow to track. It is only required in the case of “click-only” as another option of dragging a trajectory.
>
> Importantly, we also generate training data based on the same preprocess tools of SAM2. This means during training, our model also receives imperfect input of SAM2 and tries to generate results similar to the Ground Truth video. This end-to-end framework, combined with the natural denoising ability of the video diffusion model, gives our final model a certain degree of correction capability.
>
> **3. More comparison**
>
> Since the code for several recent image-conditioned video object insertion methods (e.g., VideoAnydoor, DynVFX, UNIC) has not yet been released, we compare our approach with the most recent open-source method, VACE, which introduces object ID from a given image, to strengthen our results. Corresponding results have been highlighted in red and updated in Tab.2 and Fig.7.
>
> It is important to note that existing object insertion tasks differ from our setting: they generally focus on inserting object ID from a static image and generating realistic dynamics. In contrast, our task is designed to faithfully inherit both object ID and the original foreground dynamics.

---

> ### Author Response · Authors · 2025-11-27
>
> Dear reviewer, thank you very much for your thoughtful and constructive comments. We have carefully revised the manuscript and addressed each of your points in detail. Your feedback has greatly helped improve the clarity and quality of our work. If your have any further concerns, please let us know and we will try our best to address any remaining concerns. Thank you again for your time, effort, and professional advice on our manuscript.

---

### Author Response · Authors · 2025-12-02
**Unified Response**

Dear AC and reviewers:

We appreciate the time you dedicated to reviewing our work.

We initially received a mixed score. After discussion, both reviewers who scored 4 indicated a preference to raise (R``#ThTu``) or adjust (R``#irsQ``) their final rating, but due to information leak, they did not directly change their scores (as mentioned by R``#ThTu``). We have carefully revised the manuscript and marked the changes in red. R``#irsQ`` believes that it is significantly stronger than original version.

Based on the latest ICLR principles, we earnestly request the AC to consider our discussions. We thank all the reviewers and the AC again for their hard work and time regarding our manuscript.

Best Regards,

Authors of Submission 1570

---

### Meta-Review · Area_Chair_p4er · 2026-01-05

**Summary:**

sd3W: (1) clarification of ERoPE, (2) dependency on SAM2 and optical flow, and sensitivity, (3) lack of comparisons to other methods.
ThTu:(1) not a fundamental breakthrough, (2) more elaboration of RoPE, (3) suggestion to revise Fig.3A, (4) difference from VACE, (5) adherence to user provided trajectory.
irsQ: (1) limited novelty, (2) inadequate experiments, (3) dataset concerns, (5) problem formulation/scoping.
7r6X: mostly clarification questions.

**Reviewer Concerns:**

Most of the concerns are addressed by the reviewers.
irsQ has remaining concerns on the novelty of modeling ideas, but still lean positive.

**Reviewer Scores:**

sd3W and 7r6X would have kept their scores.
ThTu and irsQ most likely would increase scores to marginal accept.

---

### Decision · Program_Chairs · 2026-01-26

Accept (Poster)